# Kap-β2/Transportin mediates β-catenin nuclear transport in Wnt signaling

Woong Y Hwang[1], Valentyna Kostiuk[1], Delfina P González[1], C Patrick Lusk[2]*, Mustafa K Khokha[1]*

[1]Pediatric Genomics Discovery Program, Department of Pediatrics and Genetics, Yale School of Medicine, New Haven, United States; [2]Department of Cell Biology, Yale School of Medicine, New Haven, United States

**Summary** Wnt signaling is essential for many aspects of embryonic development including the formation of the primary embryonic axis. In addition, excessive Wnt signaling drives multiple diseases including cancer, highlighting its importance for disease pathogenesis. β-catenin is a key effector in this pathway that translocates into the nucleus and activates Wnt responsive genes. However, due to our lack of understanding of β-catenin nuclear transport, therapeutic modulation of Wnt signaling has been challenging. Here, we took an unconventional approach to address this long-standing question by exploiting a heterologous model system, the budding yeast *Saccharomyces cerevisiae*, which contains a conserved nuclear transport machinery. In contrast to prior work, we demonstrate that β-catenin accumulates in the nucleus in a Ran-dependent manner, suggesting the use of a nuclear transport receptor (NTR). Indeed, a systematic and conditional inhibition of NTRs revealed that only Kap104, the ortholog of Kap-β2/Transportin-1 (TNPO1), was required for β-catenin nuclear import. We further demonstrate direct binding between TNPO1 and β-catenin that is mediated by a conserved PY-NLS. Finally, using *Xenopus* secondary axis and TCF/LEF (T Cell factor/lymphoid enhancer factor family) reporter assays, we demonstrate that our results in yeast can be directly translated to vertebrates. By elucidating the nuclear localization signal in β-catenin and its cognate NTR, our study suggests new therapeutic targets for a host of human diseases caused by excessive Wnt signaling. Indeed, we demonstrate that a small chimeric peptide designed to target TNPO1 can reduce Wnt signaling as a first step toward therapeutics.

*For correspondence:
patrick.lusk@yale.edu (CPL);
Mustafa.khokha@yale.edu (MKK)

## Editor's evaluation

In Wnt/β-catenin signaling, Wnt growth factor binding to cell surface receptors results in the stabilization and nuclear translocation of the transcriptional coactivator β-catenin, but how β-catenin is translocated to the nucleus has been a longstanding problem in the field. The authors show that the yeast Kap104/mammalian TNPO1 mediates nuclear translocation of β-catenin using a conserved TNPO1 nuclear localization sequence in the C-terminal region of β-catenin, and mutation of this sequence or knockdown of TNPO1 diminishes nuclear localization and Wnt sigaling. The data demonstrate that β-catenin nuclear translocation is Ran dependent and that TNPO1 binding is a significant, although not exclusive, contributor to β-catenin translocation, and could represent a new therapeutic target.

## Introduction

Wnt signaling plays multiple roles in embryonic development. For example, Wnt signaling is critical for establishing the dorsal embryonic axis; increased Wnt signaling can lead to a secondary axis (twinning of the embryo), while depletion of a key effector of the Wnt signaling pathway, β-catenin, can lead

to a radially ventralized embryo (*Heasman et al., 1994*; *Heasman et al., 2000*; *Khokha et al., 2005*; *McMahon and Moon, 1989*; *Moon et al., 1997*; *Nishisho et al., 1991*; *Smith and Harland, 1991*; *Sokol et al., 1991*). In addition, Wnt signaling has been implicated in a variety of human diseases especially cancer (*Clevers and Nusse, 2012*; *MacDonald et al., 2009*; *Moon et al., 2004*; *Morin et al., 1997*; *Nusse and Varmus, 1982*; *Polakis, 2012*; *Wood et al., 2007*). In fact, 90% of colorectal cancers are caused by genetic alterations in Wnt pathway factors (*Cancer Genome Atlas, 2012*). Therefore, chemical inhibitors of Wnt signaling have tremendous therapeutic potential.

In Wnt signaling, β-catenin (CTNNB1) relays the message from a Wnt ligand at the plasma membrane to transcription factors in the nucleus (*MacDonald et al., 2009*; *Niehrs, 2012*). As such, its levels are kept under control through a constitutively active degradation pathway. In the presence of Wnt ligand, the degradation machinery is sequestered, and the resulting stabilization of β-catenin allows it to enter the nucleus where it drives the transcription of Wnt responsive genes (*MacDonald et al., 2009*; *Niehrs, 2012*). Despite intensive study, the mechanism of β-catenin translocation from the cytosol to the nucleus remains obscure. First, although early studies suggested that β-catenin nuclear import was energy dependent, they excluded a role for the Ran GTPase (*Fagotto et al., 1998*; *Yokoya et al., 1999*), the master regulator of the nuclear transport of most macromolecules bearing either nuclear localization signals (NLSs) and/or nuclear export signals (NES; *Wente and Rout, 2010*). Second, prior studies demonstrated that β-catenin nuclear import did not require the Kap-α/Kap-β1 (Importin α/β1) nuclear transport receptor (NTR) complex (*Fagotto et al., 1998*; *Yokoya et al., 1999*), a notion consistent with Ran independence. Finally, the structural similarity between β-catenin (*Huber et al., 1997*; *Xing et al., 2008*) and Kap-α (*Conti and Kuriyan, 2000*), both made up of armadillo-repeats, suggested that β-catenin might itself act as an NTR by directly interacting with the Phe-Gly (FG) nups responsible for selective passage across the nuclear pore complex (NPC; *Andrade et al., 2001*; *Xu and Massagué, 2004*; *Yano et al., 1994*). However, Kap-α does not directly bind to FG-nups and the evidence that β-catenin does so is controversial (*Sharma et al., 2014*; *Suh and Gumbiner, 2003*). Using a CRISPR-based screening platform, recent work provided evidence that β-catenin may be imported by the NTR, IPO11 (Importin-11); however, this mechanism appears curiously relevant only for a subset of colon cancer cells (*Mis et al., 2020*). Alternative models for β-catenin nuclear transport have also been proposed; however, proteins that directly bind β-catenin to modulate nuclear transport remain undefined (*Goto et al., 2013*; *Griffin et al., 2018*; *Komiya et al., 2014*). Thus, a complete understanding of β-catenin nuclear transport remains outstanding, leaving open a key gap in our knowledge of Wnt signaling that could otherwise be targeted for therapeutic intervention.

A major challenge with understanding the β-catenin nuclear import mechanism is the myriad of binding partners that modulate its steady-state distribution and, hence, complicate the direct interrogation of the nuclear transport step (*Fagotto, 2013*; *MacDonald et al., 2009*). Here, we take an unconventional approach and investigate β-catenin nuclear transport in a heterologous system, the budding yeast *Saccharomyces cerevisiae*. Yeast do not have a Wnt pathway or a β-catenin ortholog, which presumably emerged in metazoans with the onset of multicellularity and cell fate specialization (*Holstein, 2012*). Yeast also likely lack β-catenin binding partners and its degradation machinery and thus provide a simplified system to specifically evaluate nuclear import. Most critically, the nuclear transport system, including NTRs, NPCs, and Ran, is well conserved from yeast to human (*Malik et al., 1997*; *Wente and Rout, 2010*; *Wozniak et al., 1998*). Indeed, even NLS and NES sequences are recognized by orthologous NTRs across millions of years of evolution (*Conti and Kuriyan, 2000*; *Fontes et al., 2000*; *Kosugi et al., 2008*; *Lange et al., 2008*; *Soniat et al., 2013*). Using the yeast system as a discovery platform, we uncover a PY-NLS in β-catenin that functions in yeast, *Xenopus*, and mammalian cells by directly binding to Kap-β2/Transportin-1 (TNPO1). We further demonstrate that a small peptide based on a chimeric PY-NLS sequence can prevent Wnt signaling, opening the door for therapeutics.

## Results

### β-catenin requires a functional Ran cycle to accumulate in the nucleus of *S. cerevisiae*

Investigating the β-catenin nuclear import mechanism in yeast relies on the premise that a minimal, conserved β-catenin transport machinery exists in this organism. Therefore, we first tested whether β-catenin accumulates in the yeast nucleus. Specifically, we assessed the localization of a *Xenopus* β-catenin-GFP (xβ-catenin-GFP) in a wildtype yeast strain expressing an endogenously tagged nuclear

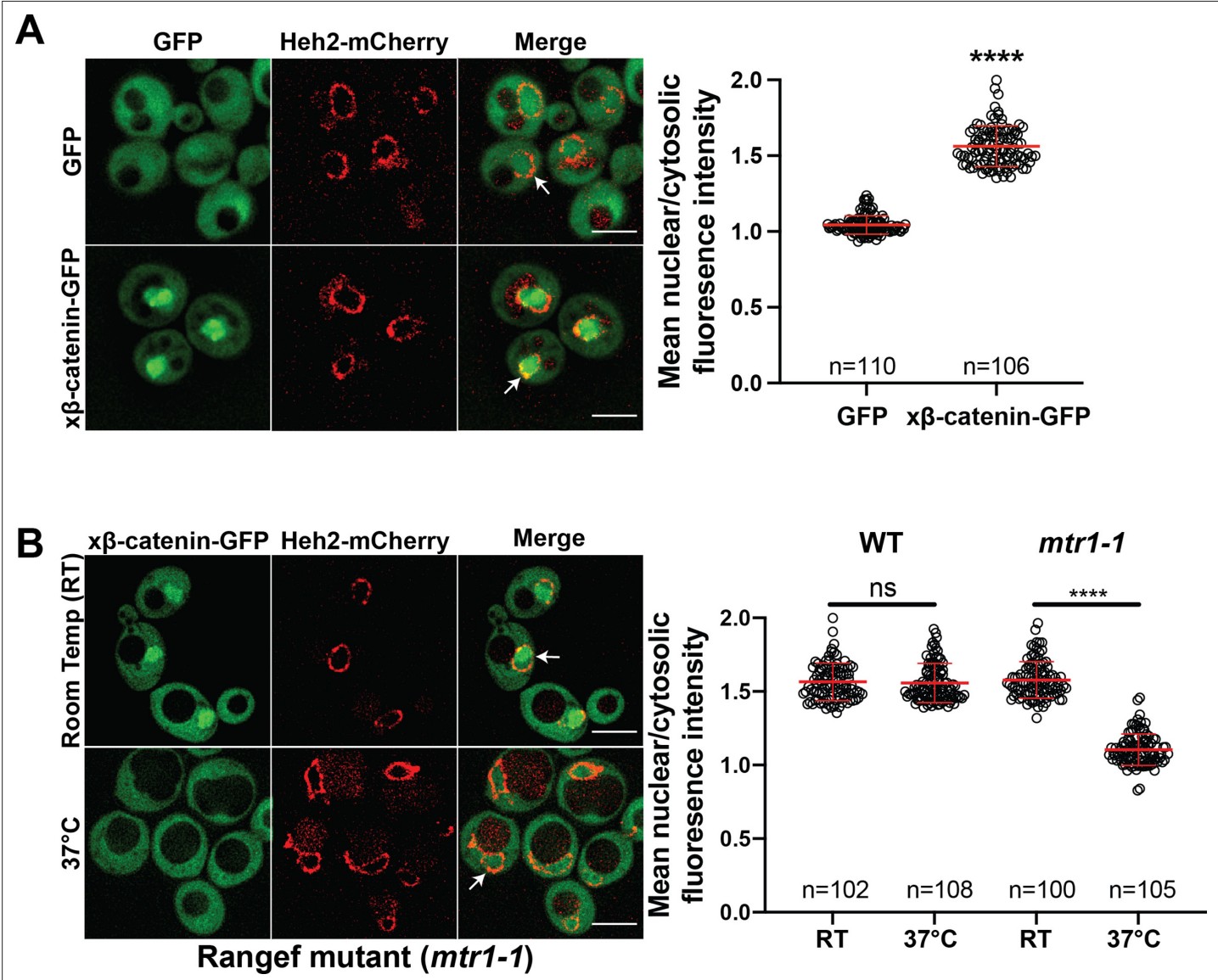

**Figure 1.** β-catenin requires a functional Ran cycle to accumulate in the nucleus of *Saccharomyces cerevisiae*. (**A**) Representative deconvolved fluorescence images of xβ-catenin-GFP in a wildtype yeast strain that expresses Heh2-mCherry to label the nucleus (left). White arrows indicate the nuclear compartment. Plot showing the quantification of mean nuclear to cytosolic fluorescence intensity from 30 to 40 cells from three independent replicates (right). (**B**) Representative deconvolved fluorescence images of xβ-catenin-GFP in the RanGEF mutant (*mtr1-1*) strain at room temperature or 37°C that co-expresses Heh2-mCherry as a nuclear envelope marker (left). The ratio of mean nuclear to cytosolic fluorescence intensity was measured in the wildtype or *mtr1-1* strain from 30 to 35 cells from three independent replicates (right). Scale bar is 5 µm in (**A**) and (**B**). Red bar indicates the mean value with the SD. p-Values are from unpaired two-tailed t-test where ns is p>0.05, and ****p<0.0001 for both (**A**) and (**B**). The data is uploaded as *Figure 1—source data 1*.

The online version of this article includes the following source data for figure 1:

**Source data 1.** Source data related to *Figure 1*.

envelope membrane protein, Heh2-mCherry, to help visualize the nuclear boundary. Indeed, xβ-catenin-GFP was enriched in the nucleus compared to GFP alone (*Figure 1A*). To quantify this steady state distribution, we measured the nuclear enrichment of xβ-catenin-GFP by relating the mean GFP fluorescence in the nucleus (N) and cytoplasm (C) (*Figure 1A*, plot at right). The xβ-catenin-GFP had a mean N:C ratio of ~1.6, which was significantly higher than GFP alone (1.1). As xβ-catenin-GFP is 119 kD, it would be unable to easily pass through the NPC diffusion barrier, suggesting that xβ-catenin-GFP can access a facilitated nuclear transport mechanism through the NPC (*Popken et al., 2015*; *Timney et al., 2016*).

In principle, three mechanisms of xβ-catenin nuclear import are possible: (1) xβ-catenin-GFP is imported by an NTR, (2) xβ-catenin-GFP piggybacks on an unknown binding partner that is itself imported by an NTR, or (3) xβ-catenin-GFP has an intrinsic ability to cross the NPC free of NTRs. To rule out the latter possibility, we tested whether xβ-catenin-GFP nuclear accumulation was dependent on a functional Ran gradient, which would specifically impact NTR-mediated transport (*Schmidt and Görlich, 2016*; *Weis, 2003*; *Wente and Rout, 2010*). We therefore assessed β-catenin-GFP localization in the *mtr1-1* mutant strain, which is a temperature sensitive, loss of function allele in the gene encoding the yeast Ran-GEF (*SRM1/MTR1/PRP20*; *Kadowaki et al., 1992*). As Ran-GEF exchanges Guanosine-5′-diphosphate (GDP) for Guanosine-5′-triphosphate (GTP) on Ran in the nucleus, it is essential for the functioning of the nuclear transport system (*Weis, 2003*). At room temperature, xβ-catenin-GFP is localized in the nucleus with a N:C ratio similar to the wildtype strain (~1.6) (*Figure 1B*). In striking contrast, growth at 37°C, which is non-permissive for mtr1-1p function, resulted in the re-distribution of xβ-catenin-GFP such that it was evenly distributed between the nucleus and cytoplasm with N:C ratios identical to GFP alone (1.1) (*Figure 1B*). Importantly, xβ-catenin-GFP localization was not affected by the elevated temperature as wildtype cells showed N:C ratios of ~1.6 even at 37°C. Thus, nuclear accumulation of xβ-catenin-GFP is dependent on a functional Ran-GTP gradient raising the possibility that it requires a NTR-mediated pathway to accumulate in the nucleus.

## The C-terminus of β-catenin contains an NLS

Having established that β-catenin import requires a functional Ran pathway, we next sought to map the sequence elements of β-catenin that confer nuclear localization. β-catenin can be divided into three domains: a central region rich in armadillo (ARM) repeats sandwiched between two unstructured domains (*Figure 2A*). We generated constructs where each of these domains was individually deleted and examined their localization in yeast. Removal of the C-terminus significantly reduced nuclear enrichment of xβ-catenin-(1-664)-GFP compared to constructs lacking either the N (xβ-catenin-[141-782]-GFP) or ARM (Δ141-664) domains, which accumulated in the nucleus at levels similar to the full-length protein (*Figure 2A–C*). These data suggested that the C-terminus contains sequence elements required for nuclear accumulation. Consistent with this idea, the C-terminus of β-catenin (xβ-catenin-[665-782]-GFP) was sufficient to confer nuclear accumulation of GFP to levels comparable to the full-length protein (mean N:C 1.6; *Figure 2A, B and D*). Of note, both the ARM repeats (xβ-catenin-[141-664]-GFP) and the N-terminus of β-catenin (xβ-catenin-[1-141]-GFP) could confer some nuclear enrichment of GFP but to a considerably lesser extent than the C-terminus (mean N:C of ~1.3; *Figure 2A, B and D*). Additionally, the ARM repeats had some affinity for the nuclear periphery (*Figure 2*; xβ-catenin-[141-664]-GFP, white arrows). Of future interest, even the weaker nuclear accumulation driven by the N-terminal and ARM repeat sequences is dependent on a functional Ran pathway as xβ-catenin-1–664 does not accumulate in the nucleus in the *mtr1-1* strain at the non-permissive temperature (*Figure 2—figure supplement 1*). Thus, when taken together, there are several elements of xβ-catenin that, in isolation, can target to the nucleus in a Ran pathway dependent fashion, but the C-terminus contains a sequence that was both necessary and sufficient for nuclear accumulation at levels comparable to the full-length protein. Consistent with previous work (*Koike et al., 2004*; *Mis et al., 2020*), these data suggested that the C-terminus of xβ-catenin contains the dominant NLS in β-catenin, which we further mapped to amino acids 665–745 (*Figure 2A, B and E*).

We confirmed that nuclear accumulation of xβ-catenin-(665-745)-GFP was dependent on the Ran pathway using the *mtr1-1* strain (*Figure 2—figure supplement 2A*). To ensure that this sequence did not confer binding to a yeast-specific factor, we also tested localization of xβ-catenin-(665-745)-GFP in HEK293T cells, a human embryonic kidney cell line. In line with the yeast results, xβ-catenin-(665-745)-GFP showed higher levels of nuclear accumulation compared to GFP alone (*Figure 2—figure*

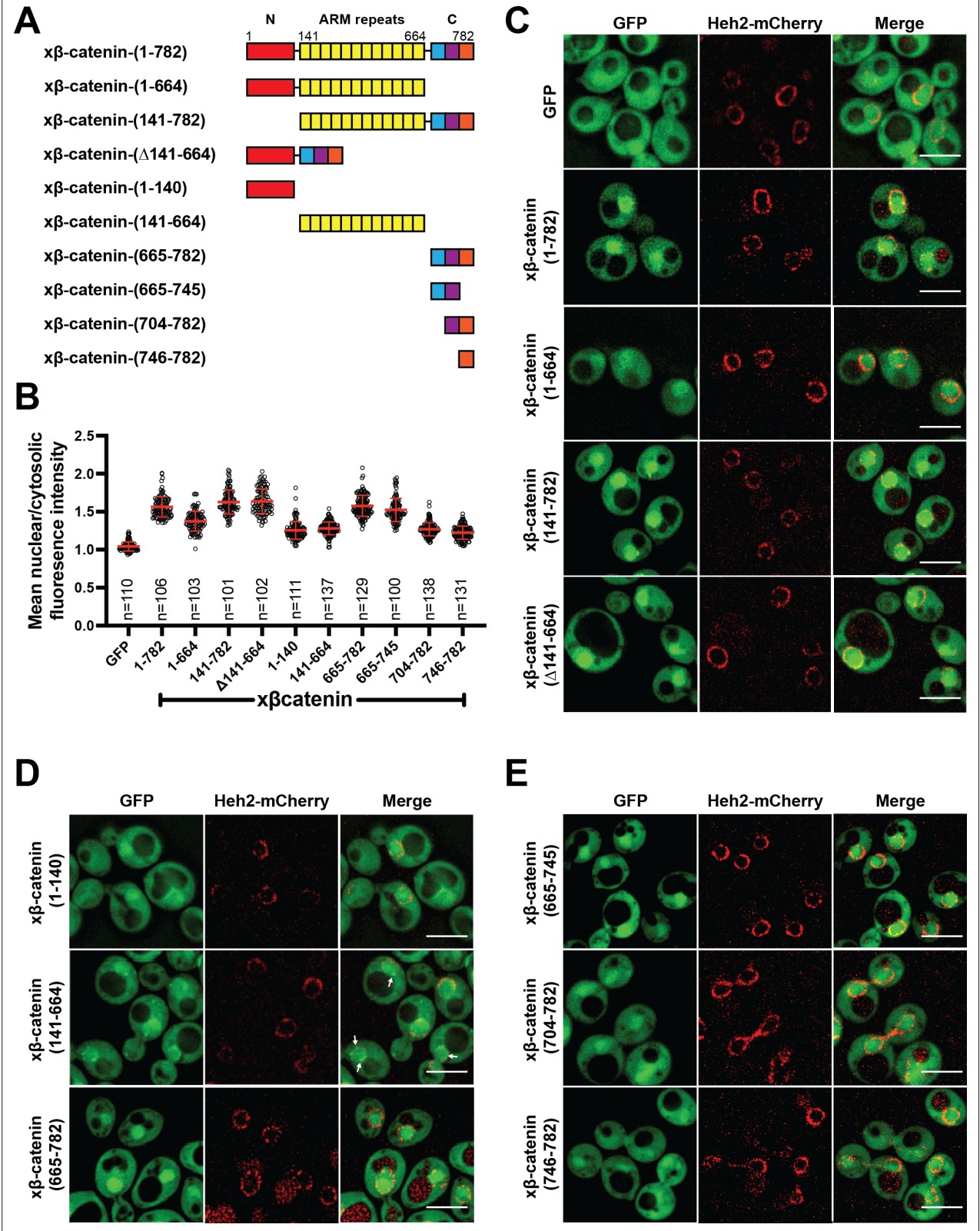

**Figure 2.** The C-terminus of β-catenin contains a nuclear localization signal (NLS). (**A**) Schematic of *Xenopus* β-catenin truncation constructs tested in this study. (**B**) Plot of the ratio of mean nuclear to cytosolic fluorescence intensity of *Xenopus* β-catenin GFP truncation constructs tested in a wildtype yeast strain from 30 to 40 cells from three independent replicates. Red bar indicates the mean value with the SD. (**C**) Deconvolved fluorescence images of the N-terminal deletion (141-782), ARM-repeats deletion (Δ141-664) and C-terminal deletion (1-664) of *Xenopus* β-catenin GFP in the wildtype strain.

*Figure 2 continued on next page*

*Figure 2 continued*

GFP and full-length *Xenopus* β-catenin-GFP were used as controls. (**D**) Deconvolved fluorescence images of the indicated fragments of *Xenopus* β-catenin GFP in the wildtype strain. White arrows indicate nuclear rim localization. (**E**) Deconvolved fluorescence images of indicated C-terminus fragments of *Xenopus* β-catenin GFP in the wildtype strain. Heh2-mCherry was co-expressed to label the nuclear envelope in (**C**), (**D**), and (**E**). Scale bar is 5 µm in (**C**), (**D**), and (**E**). The data is uploaded as *Figure 2—source data 1*.

The online version of this article includes the following source data, source code, and figure supplement(s) for figure 2:

**Source data 1.** Source data related to *Figure 2*.

**Figure supplement 1.** β-catenin (1-664) localizes to the nucleus in a RanGTPase dependent manner in *Saccharomyces cerevisiae*.

**Figure supplement 1—source code 1.** Source data related to *Figure 2—figure supplement 1*.

**Figure supplement 2.** Ran dependence of the β-catenin (665-745)-GFP in the *Saccharomyces cerevisiae* and its requirement for nuclear localization in HEK293T cells.

**Figure supplement 2—source data 1.** Source data related to *Figure 2—figure supplement 2*.

**Figure supplement 3.** Residues 665–745 of β-catenin are required to induce secondary axes in *Xenopus laevis.*

*supplement 2B*). Furthermore, to test if this NLS is required for β-catenin localization in human cells, we also examined human β-catenin that lacks aa 665–745 in HEK293T cells. Indeed, deletion of aa 665–745 (hβ-catenin-[Δ665–745]-GFP) led to a significant reduction in nuclear to cytoplasmic ratio compared to the full-length hβ-catenin localization (*Figure 2—figure supplement 2C*).

Next, we investigated the function of the β-catenin NLS in the context of Wnt signaling using the secondary axis assay in *Xenopus* (*McMahon and Moon, 1989*; *Smith and Harland, 1991*; *Sokol et al., 1991*). Overexpression of Wnt effectors including β-catenin induces a secondary axis in *Xenopus* embryos. By injecting a moderate dose (200 pg) of xβ-catenin mRNA, secondary axes develop in roughly half of the embryos (*Figure 2—figure supplement 3A-B*) compared to none in the uninjected controls (UICs). If we delete the coding sequence for the NLS (xβ-catenin-[Δ665–745]-GFP) then the number of embryos with secondary axes is significantly reduced (*Figure 2—figure supplement 3A-B*). To ensure that this loss of function is due to the inhibition of β-catenin nuclear import, we added the classical NLS of the SV40 large T-antigen (cNLS), which is imported by Kap-α/β1, to the N-terminus of this construct (cNLS-xβ-catenin-[Δ665–745]-GFP). cNLS-xβ-catenin-(Δ665–745)-GFP could induce secondary axes similarly to the full-length β-catenin (*Figure 2—figure supplement 3B*), consistent with the conclusion that xβ-catenin-(Δ665–745)-GFP was still functional for Wnt signaling but lacked the nuclear localization element. Supporting this supposition, we imaged xβ-catenin-GFP in these *Xenopus* embryos and found that while xβ-catenin-GFP localizes to the cell membrane and nucleus, xβ-catenin-(Δ665–745)-GFP shows a reduction in nuclear accumulation, which is rescued by the addition of the cNLS (*Figure 2—figure supplement 3C*).

## Kap104 is specifically required for β-catenin nuclear accumulation in *S. cerevisiae*

Next, to define the NTR responsible for xβ-catenin-GFP nuclear import, we used the Anchor-Away approach (*Haruki et al., 2008*) to systematically inhibit 10 budding yeast NTRs, all of which have orthologs in human cells (*Table 1*). This strategy takes advantage of the rapamycin-induced dimerization of a FK506 binding protein (FKBP12) with the FKBP-rapamycin binding (FRB) domain (*Figure 3A*). In this system, NTR-FRB fusions are expressed in a strain harboring FKBP12 fused to

**Table 1.** List of human nuclear transport receptor (NTR) and *Saccharomyces cerevisiae* orthologs.

| Human NTR | *S. cerevisiae* orthologs |
| --- | --- |
| Kap β1 | Kap95 |
| Kap α | Kap60/Srp1 |
| Transportin 1 | Kap104 |
| Importin-5/Kap β3 | Kap121/Pse1 |
| Importin-4/RanBP5 | Kap123 |
| Importin-7/RanBP7 | Kap119/Nmd5 |
| Importin-8/RanBP8 | Kap108/Sxm1 |
| Importin-9 | Kap114 |
| Importin-11 | Kap120 |
| Transportin-SR/TNPO3 | Kap111/Mtr10 |
| Importin-13 | Kap122/Pdr6 |
| CRM1/Exportin-1 | Xpo1 |
| Exportin-t | Los1 |
| Exportin-5 | Kap142/Msn5 |
| CAS | Cse1 |

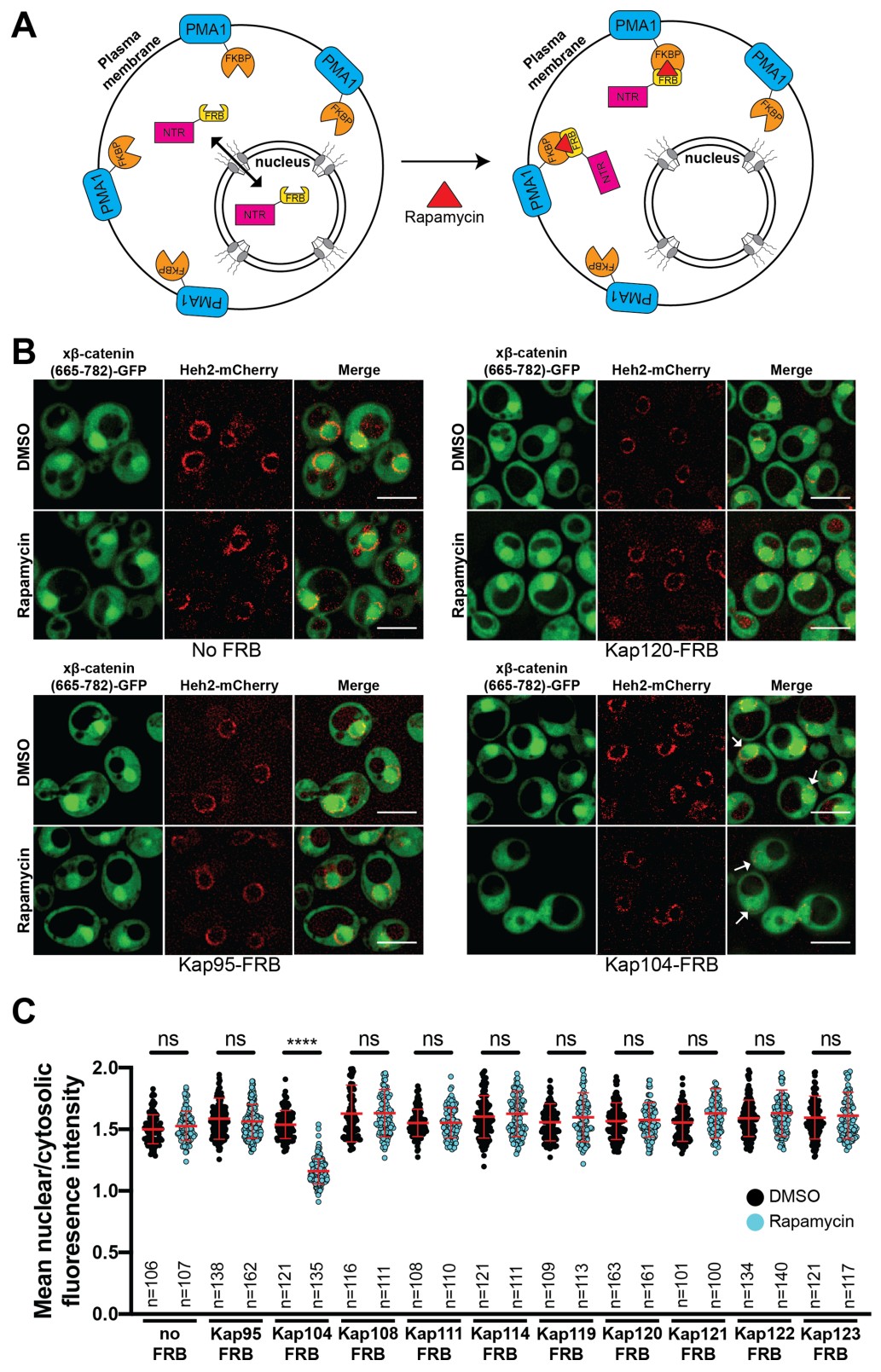

**Figure 3.** Kap104 is specifically required for β-catenin nuclear accumulation in *Saccharomyces cerevisiae*. (**A**) Schematic of the Anchor-Away assay mediated by the rapamycin-induced dimerization of nuclear transport receptor (NTR)-FKBP-rapamycin binding (FRBP and Pma1-FKBP12). Pma1 is a plasma membrane ATPase. (**B**) Deconvolved fluorescence images of cells with indicated FRB fusions expressing *Xenopus* β-catenin (665-782)-GFP

*Figure 3 continued on next page*

*Figure 3 continued*

treated with DMSO (vehicle) or rapamycin for 15 min. Heh2-mCherry was used as a nuclear envelope marker. White arrows indicate the nucleus. Scale bar is 5 μm. (**C**) Plot showing the ratio of mean nuclear to cytosolic fluorescence intensity of *Xenopus* β-catenin (665-782)-GFP in the 10 NTR-FRB strains treated with DMSO or rapamycin from 30 to 40 cells from three independent replicates. Red bar indicates the mean value with the SD. Experiments were performed three times. p-Values are from unpaired two-tailed t-test where ns is p>0.05 and ****p<0.0001. The data is uploaded as *Figure 3—source data 1*.

The online version of this article includes the following source data and figure supplement(s) for figure 3:

**Source data 1.** Source data related to *Figure 3*.

**Figure supplement 1.** Sub-cellular localization of *Xenopus* β-catenin (665-782)-GFP in Anchor-Away strains in *Saccharomyces cerevisiae.*

**Figure supplement 2.** Anchor-Away cloning strategy in *Saccharomyces cerevisiae.*

a highly abundant plasma membrane protein (Pma1) (*Figure 3A*). The addition of rapamycin leads to the rapid (~15 min) trapping of the NTRs at the plasma membrane (*Haruki et al., 2008*). We systematically tested whether the addition of rapamycin (or the DMSO carrier alone) impacted the nuclear accumulation of xβ-catenin-(665-782)-GFP in each of the 10 NTR-FRB strains. Consistent with prior data (*Fagotto et al., 1998*), plasma membrane trapping of the Kapβ1 ortholog, Kap95-FRB, did not impact nuclear localization of xβ-catenin-(665-782)-GFP (*Figure 3B*). Indeed, trapping 9 of the 10 NTRs, including the IPO11 ortholog, Kap120, had no overt influence on xβ-catenin-(665-782)-GFP nuclear localization (*Figure 3B and C*, and *Figure 3—figure supplement 1*). In contrast, we observed a remarkable inhibition of nuclear accumulation, specifically when Kap104-FRB (ortholog of Kapβ2/Transportin-1[TNPO1]) was anchored away (*Figure 3B and C*). These data support a model in which Kap104 specifically mediates the nuclear import of xβ-catenin-(665-782)-GFP in the yeast system.

## β-catenin contains a PY-NLS that is required for nuclear import

Having established that Kap104 mediates β-catenin nuclear transport in yeast, we compared the xβ-catenin-(665-782) protein sequence to established TNPO1 NLS (e.g. PY-NLS; *Lee et al., 2006*; *Soniat and Chook, 2015*; *Soniat et al., 2013*). By close inspection, the xβ-catenin amino acid sequence (665-703) does conform to the loose PY-NLS consensus with a hydrophobic methionine (M) in place of a tyrosine (Y; *Figure 4A*; *Soniat and Chook, 2015*); this NLS is conserved across vertebrate species (*Figure 4A*). We therefore mutated the PM motif in β-catenin by substituting the proline (P) and methionine (M) residues for tandem alanine (A) amino acids. We tested whether these changes impacted the ability of the xβ-catenin-(665-703) to import a GFP fusion to three maltose binding proteins. MBP(x3)-GFP is constitutively excluded from the nucleus due to its large molecular weight (149 kDa) (*Figure 4B*; *Popken et al., 2015*). Fusion of the xβ-catenin-(665-703) can confer nuclear localization of this large fusion protein. Furthermore, this localization is dependent on the PM motif as substitution of PM with AA abolishes nuclear localization (*Figure 4B*). These amino acids were also critical for human (h)β-catenin-(665-782)-GFP nuclear accumulation in human cell lines (HeLa) as the PM to AA substitution reduced the mean N:C ratios of this construct from 2.5 to 1.5 (*Figure 4C*). Thus, the PM sequence in the β-catenin PY-NLS is required for nuclear import in yeast and human cells.

## Direct binding of β-catenin and TNPO1 is destabilized by Ran-GTP

To test that the β-catenin-NLS is directly recognized by TNPO1, we generated recombinant TNPO1 and Glutathione S-transferase (GST) fusions of human β-catenin (GST-hβ-catenin) and human β-catenin containing the PM-AA mutations (GST-hβ-catenin P687A, M688A). We immobilized these GST fusions (and GST alone) on GT Sepharose beads and tested binding to purified TNPO1. We observed specific binding of TNPO1 to the GST-hβ-catenin (*Figure 5A*), which was disrupted by the PM to AA mutations in the NLS. Furthermore, as NTR-NLS interactions are disrupted by the binding of Ran-GTP to the NTR in the nucleus, we tested the Ran-GTP sensitivity of the TNPO1:β-catenin complex. We generated recombinant RanQ69L, which cannot hydrolyze GTP and confirmed that RanQ69L-GTP specifically interacted with TPNO1 and not to β-catenin (*Figure 5—figure supplement 1B*). Next, we assessed the formation of the TNPO1:β-catenin complex in the presence or absence of RanQ69L-GTP. Adding RanQ69L-GTP specifically disrupted β-catenin binding to TPNO1, indicating that the TNPO1:β-catenin

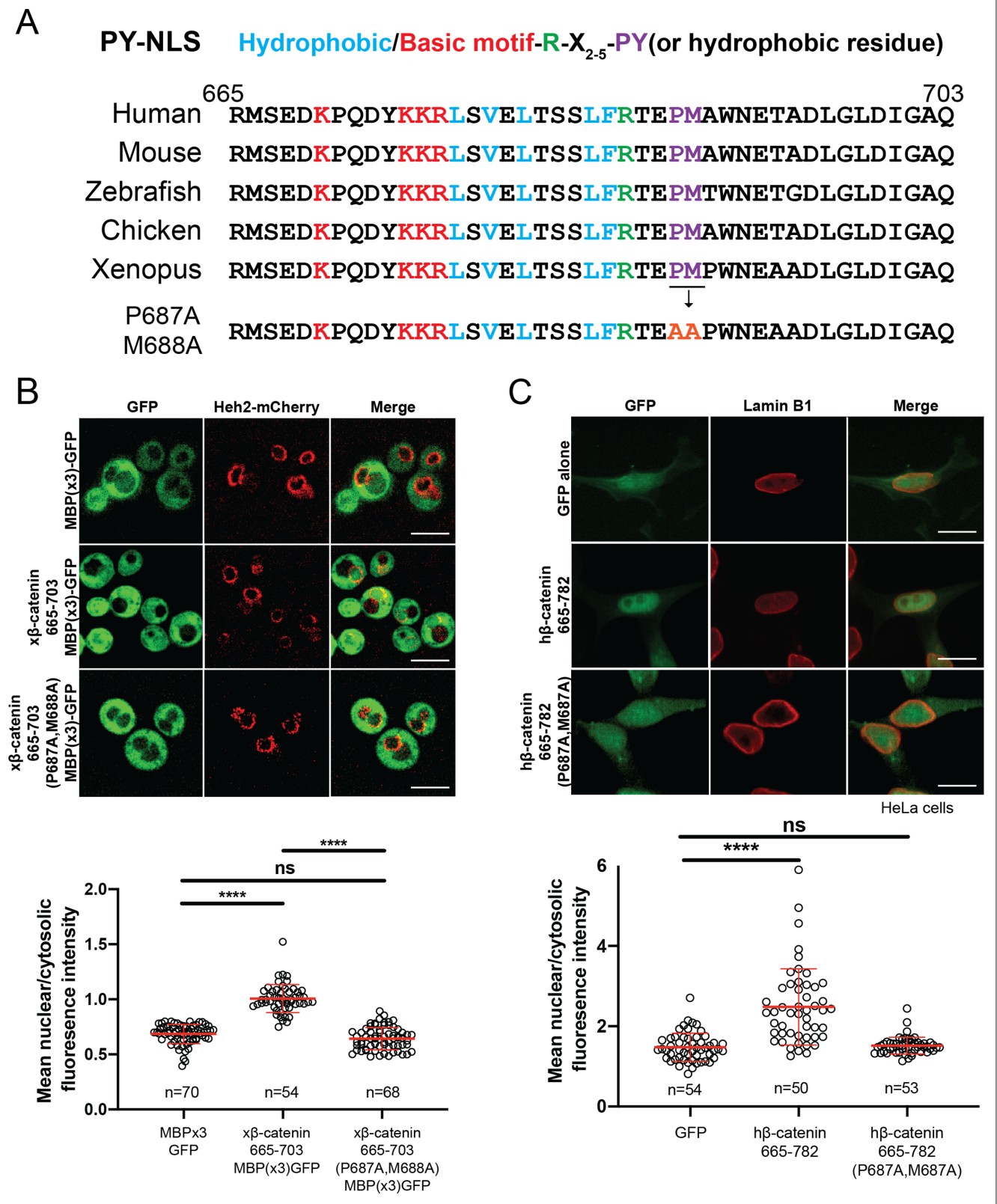

**Figure 4.** β-catenin contains a PY-NLS that is required for nuclear import. (**A**) Conservation of amino acid sequences that conform to the PY-NLS consensus (top, from *Soniat and Chook, 2015*) in the C-terminus of β-catenin. (**B**) Deconvolved fluorescence images of wildtype yeast cells expressing MBP(x3)-GFP tagged with the *Xenopus* β-catenin nuclear localization signal (NLS) (665-703) and also an NLS that contains the PM to AA mutation (top). Untagged MBP(x3)-GFP was used as a control. Plot of the ratio of mean nuclear to cytoplasmic fluorescence intensity from a single experiment (bottom).

*Figure 4 continued on next page*

*Figure 4 continued*

Scale bar is 5 µm. (**C**) Representative fluorescence image of HeLa cells expressing human β-catenin (665-782) or the PM to AA mutant version (top). LaminB1 was labeled to locate the nuclear envelope. GFP alone was used as a control. Plot of the ratio of mean nuclear to cytoplasmic fluorescence intensity from three experiments (bottom). Scale bar is 15 µm. p-Values are from unpaired two-tailed t-test where ns is p>0.05 and ****p<0.0001 for both (**B**) and (**C**). The data is uploaded as *Figure 4—source data 1*.

The online version of this article includes the following source data for figure 4:

**Source data 1.** Source data related to *Figure 4*.

complex reflects the formation of a canonical NTR-NLS import complex (*Figure 5B*). When taken together, these data establish a model in which TNPO1 imports β-catenin through a direct interaction with its PY-NLS that is modulated by TPNO1-binding to Ran-GTP.

## TNPO1/2 and the β-catenin NLS are required for Wnt signaling in vivo

To explore the function of TNPO1-mediated import of β-catenin in vertebrates, we applied two different Wnt signaling assays: (1) a TCF/LEF reporter and (2) *Xenopus* secondary axis development. First, once β-catenin enters the nucleus, it binds to the TCF/LEF complex to activate transcription of Wnt responsive genes (*Molenaar et al., 1996*; *van de Wetering et al., 1997*; *van de Wetering et al., 1991*). A well-established reporter assay (commonly known as TOPFLASH) places the TCF/LEF DNA-binding element upstream of a reporter such as GFP or luciferase (*Molenaar et al., 1996*; *van de Wetering et al., 1991*). In *Xenopus tropicalis*, the *Tg(pbin7Lef-dGFP)* line has seven tandem TCF/LEF DNA binding sites upstream of GFP and is an effective reporter of Wnt signaling (*Borday et al., 2018*; *Denayer et al., 2008*). We crossed heterozygous transgenic animals with a wildtype animal such that half of the resultant progeny had the transgene. In vertebrates, the *tnpo1* gene is duplicated (*tnpo1* and *tnpo2*), and both paralogs have nearly identical sequence and function (*Figure 6—figure supplement 1*; *Dormann et al., 2010*; *Rebane et al., 2004*; *Twyffels et al., 2014*). Therefore, we injected single guide RNAs (sgRNAs) targeting both *tnpo1* and *tnpo2* with Cas9 protein at the one cell stage and raised embryos to st10 before fixing them. Because GFP fluorescence is undetectable at these early stages, we used whole mount in situ hybridization (WMISH) to visualize GFP transcripts as an assay for Wnt reporter activation and used sibling embryos without the transgene as a WMISH negative control. When we depleted both *tnpo1* and *tnpo2* using F0 CRISPR, significantly more embryos had weak expression of the GFP transgene compared to UIC embryos (*Figure 6A*, see images of embryos stained for GFP transcripts as key to histogram). This result was specific as the second set of non-overlapping sgRNAs gave comparable results (*Figure 6A*, *tnpo1/2* sgRNAs#2). Importantly, we detected deleterious gene modification at the appropriate targeted sites using Inference of CRISPR Edits (ICE) analysis (*Figure 6—figure supplement 2*).

We next tested the function of mouse Tnpo1/2 in a stable transgenic mouse fibroblast cell line in which luciferase is expressed under the control of TCF/LEF DNA-binding elements. To activate Wnt signaling, we transfected a full-length human β-catenin-GFP that increased luciferase signal 7.7-fold over transfection of GFP alone (*Figure 6B*). Then we measured luciferase activity in transgenic fibroblasts transfected with hβ-catenin-GFP in which we depleted transcripts of TNPO1 or TNPO2 (alone or simultaneously) using specific siRNAs. Compared to control siRNA, depletion of either TNPO1 or TNPO2 led to a 34% reduction in luciferase signal (*Figure 6B*). By targeting both transcripts, the luciferase signals were reduced by 64% (*Figure 6B*). By western blot, we observed the production of hβ-catenin-GFP and the specific reduction of TNPO1/2 (*Figure 6—figure supplement 3*). We next examined the dependence of β-catenin nuclear localization on TNPO1/2 in colorectal cancer cell lines, HCT-116 and DLD-1, which harbor an activating Ser45 β-catenin mutation and a deactivating frameshift mutation in APC, respectively. Similar to the TCF/LEF mouse fibroblast cell line, TNPO1/2 depletion by siRNA reduced β-catenin levels specifically in nuclear fractions compared to control siRNA condition (*Figure 6—figure supplement 4*). Taken together, TNPO1/2 is required for β-catenin nuclear localization across a wide range of species: *Xenopus*, mouse, and human cancer cell lines.

Having established the importance of TNPO1/2 for β-catenin nuclear localization in vertebrates, we next evaluated the specific impact of inhibiting β-catenin nuclear import by testing the function of the PY-AA mutant in the *Xenopus* secondary axis assay. We compared the number of secondary axes induced by the wildtype hβ-catenin mRNA to a PM to AA mutated version (P687A, M688A). We noted a significant reduction in the number of secondary axes induced by the PM-AA mutant

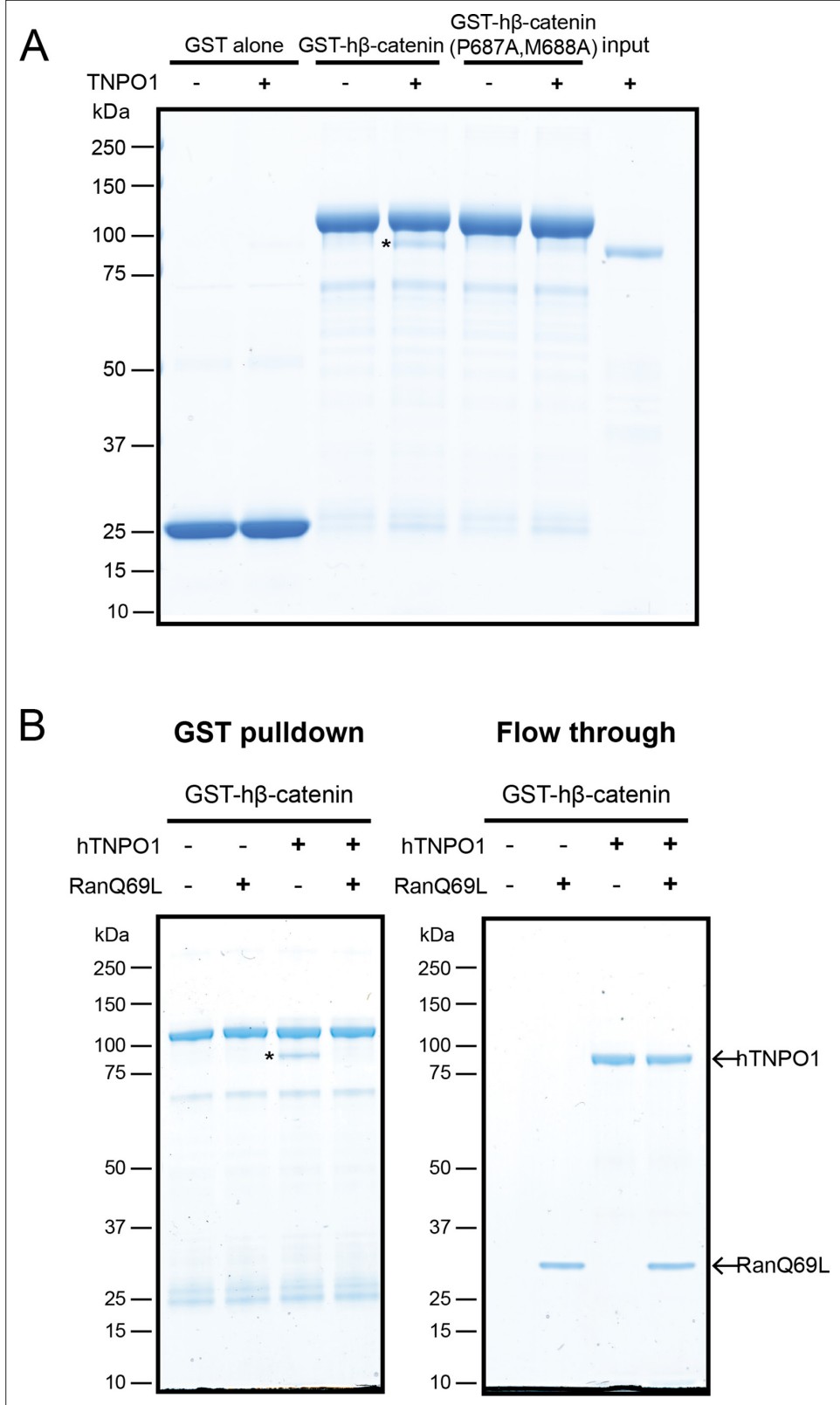

**Figure 5.** Direct binding of β-catenin and TNPO1 is destabilized by Ran-GTP. (**A**) In vitro binding assay of purified recombinant TNPO1 and GST fusions of human β-catenin and human β-catenin containing the PM to AA mutations. GST alone was used as a negative control. (**B**) In vitro binding assay of purified recombinant TNPO1 to GST fusions of human β-catenin in the presence of GTP hydrolysis deficient Ran mutant loaded with GTP

*Figure 5 continued on next page*

*Figure 5 continued*
(RanQ69L). Proteins were separated by sodium dodecyl sulfate–polyacrylamide gel electrophoresis (SDS-PAGE) and stained with Coomassie blue in (**A**) and (**B**). * indicates TNPO1 bound to GST-hβ-catenin.

The online version of this article includes the following figure supplement(s) for figure 5:

**Figure supplement 1.** TNPO1 selectively binds to RanGTP in vitro.

β-catenin (*Figure 6C*). If we add the cNLS to the N-terminus of the PM-AA mutant then induction of secondary axes is rescued, suggesting that the PM-AA mutant fails to enter the nucleus to activate Wnt signaling (*Figure 6C*).

## The M9M peptide inhibits Wnt signaling

Having established that TNPO1 binds directly to a PY-NLS and imports β-catenin into the nucleus, we wondered whether direct perturbation of the β-catenin-TNPO1 interaction could, in principle, be a viable therapeutic strategy. We therefore took advantage of the prior design of a potent TNPO1 peptide inhibitor, M9M, that binds with high affinity to the TNPO1 NLS binding site (*Cansizoglu et al., 2007*). We tested whether this peptide could inhibit Wnt signaling in the mouse fibroblast TCF/LEF luciferase reporter cell line (*Cansizoglu et al., 2007*). Excitingly, transfection of the M9M peptide reduced luciferase activity in a dose-dependent manner, regardless of whether activation was induced with a Wnt ligand (Wnt3a) or by co-transfection with human β-catenin (*Figure 7A*, *Figure 7—figure supplement 1A-B*).

The M9M peptide is a chimera of the NLSs of heterogeneous nuclear ribonucleoprotein (hnRNP) M and A1 (*Cansizoglu et al., 2007*). To test the specificity, we leveraged the understanding of the key amino acids that confer binding to TNPO1 in each individual NLS to design a control M9M-A peptide. M9M-A contains seven amino acid substitutions that would reduce binding to TNPO1 (*Figure 7—figure supplement 1C*). The M9M-A peptide only reduced luciferase activity by 24% (compared to 45% by M9M at the similar dosage; *Figure 7B*).

As the M9M peptide inhibits the nuclear import of a multitude of TNPO1/2 cargos (*Cansizoglu et al., 2007*), we sought to ensure that the M9M-mediated inhibition of Wnt signaling was specifically due to the reduced nuclear import of β-catenin. We therefore transfected a human β-catenin with a cNLS, which would be imported by Kap-α/Kap-β1 and thus be insensitive to M9M inhibition. The cNLS-human β-catenin could drive the luciferase reporter to levels comparable to human β-catenin, but the M9M peptide only reduced this signal by ~15% (*Figure 7B*). Thus, these data support the conclusion that the M9M peptide's impact on Wnt signaling is due, at least in part, to the inhibition of β-catenin nuclear transport via TNPO1/2. Together, Wnt signaling can be inhibited by blocking TNPO1-mediated import of β-catenin either by mutating the β-catenin NLS or competitive inhibition with the M9M peptide.

## Discussion

Our lack of understanding of the mechanism of β-catenin nuclear transport has been a major stumbling block for opening up new avenues of Wnt-targeted anti-cancer therapies. Numerous transport models have been proposed including piggybacking on TCF/LEF (*Behrens et al., 1996*; *Huber et al., 1996*; *Molenaar et al., 1996*) or APC (*Henderson, 2000*; *Neufeld et al., 2000*; *Rosin-Arbesfeld et al., 2000*), although neither factor was ultimately found to be required for import (*Eleftheriou et al., 2001*; *Orsulic and Peifer, 1996*; *Prieve and Waterman, 1999*). There is also a model in which β-catenin directly binds to the FG-nups and translocates through the NPC in an NTR-like (yet Ran independent) mechanism (*Fagotto, 2013*; *Xu and Massagué, 2004*). Indeed, while some of our data suggests that there are additional nuclear localization elements in the ARM repeats or N-terminus (*Figure 2—figure supplement 1*) that might also confer an affinity for the NPC (*Figure 2D*, white arrows), they are dependent on a functional Ran cycle. Thus, we disfavor the idea that there is a Ran-independent pathway for β-catenin nuclear import while acknowledging that there is still more to be learned.

By far the major determinant of β-catenin's nuclear accumulation is the PY-NLS in its C-terminus. This NLS functions in yeast, *Xenopus*, and human cells, supporting that it can be recognized by TPNO1 orthologs across eukaryotes. It turns out, however, that although the β-catenin NLS can be imported

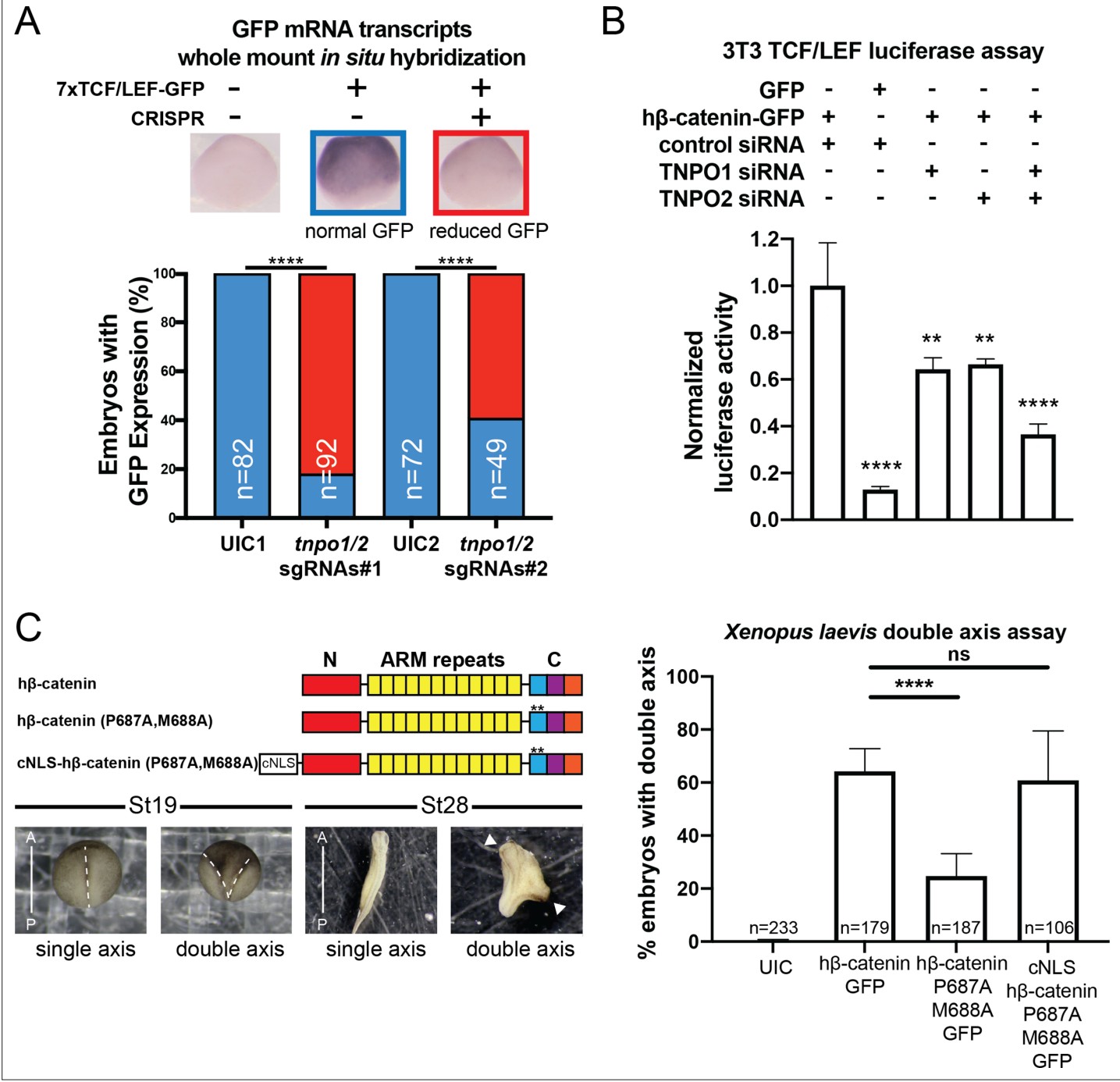

**Figure 6.** TNPO1/2 and the β-catenin nuclear localization signal (NLS) are required for Wnt signaling in vivo. (**A**) Depletion of tnpo1 and tnpo2 using two different pairs of non-overlapping sgRNAs represses gfp expression in *Xenopus tropicalis* Tg(pbin7Lef-dGFP) embryos at stage 10. Key used to quantify embryos with whole mount in situ hybridization (WMISH) signal (blue – normal gfp signal nd red – reduced gfp signal). Uninjected control (UIC) embryos were used as a negative control. (**B**) siRNA mediated TNPO1 and/or TNPO2 knockdown reduces luciferase activity in mouse embryonic fibroblasts that harbor a stable integration of luciferase under the control of TCF/LEF promoters. Wnt signaling was activated by human β-catenin-GFP overexpression. Control siRNA and GFP were used as negative controls. Experiments were performed in triplicate. (**C**) Schematic diagram of three β-catenin constructs used in the double axis assay in *Xenopus laevis*. ** indicates P687A, M688A substitutions (top left). Dorsal views of *X. laevis* embryos with anterior to the top (bottom left). Dotted lines indicate the embryonic axis, and the white arrows indicate the head. Histogram of the percent of embryos with secondary axes from three independent replicates. p-Values are from Fisher's exact test (**A**) and (**C**) and unpaired two-tailed t-test (**B**) where ns is p>0.05, p<0.05 (*), 0.0021 (**), 0.0002 (***), and p<0.0001 (****).The data is uploaded as *Figure 6—source data 1*.

The online version of this article includes the following source data and figure supplement(s) for figure 6:

*Figure 6 continued on next page*

*Figure 6 continued*

**Source data 1.** Source data related to *Figure 6*.

**Figure supplement 1.** Sequence alignment of Tnpo1 and Tnpo2 across species.

**Figure supplement 2.** *X. tropicalis tnpo1* and *tnpo2* gene depletion by CRISPR/Cas9.

**Figure supplement 3.** Western blots of Tnpo1/2 and β-catenin from 3T3 TCF/LEF luciferase assays.

**Figure supplement 4.** TNPO1/2 regulates nuclear β-catenin levels in colorectal cancer cells.

in yeast, Kap104 (the yeast TPNO1 ortholog) does not recognize the same spectrum of NLSs as its human counterpart (*Süel et al., 2008*). Indeed, Kap104 only recognizes PY-NLSs with an N-terminal basic motif, whereas human TPNO1 can bind to PY-NLSs with both a basic or a hydrophobic N-terminal motif (*Soniat and Chook, 2016*; *Süel et al., 2008*). Thus, we were fortunate that β-catenin's PY-NLS could be recognized by Kap104, enabling our yeast screen and the ultimate identification of the dominant NLS in β-catenin.

This work must also be reconciled with a recent study that implicated IPO11 as an important factor for β-catenin nuclear transport in a subset of cancer cells (*Mis et al., 2020*). The latter work also identified the C-terminus as essential for β-catenin nuclear transport, and our study adds further resolution to a specific PY-NLS. However, we did not determine any requirement for IPO11 (Kap120) in β-catenin import, at least in the yeast system. Although it remains possible that IPO11 might ultimately have a role in β-catenin nuclear transport in some vertebrate cells, the current evidence suggests that it plays a critical role only in a subset of colon cancer cells (*Mis et al., 2020*). In contrast, our work establishes that TNPO1 imports β-catenin in yeast, amphibian, mouse, and human cells, providing confidence that there is a TNPO1-specific mechanism at play. How IPO11 contributes to β-catenin nuclear import in the context of specific cancer cell lines remains to be fully established. It also remains possible that there are additional NLSs in β-catenin as the N-terminus and ARM repeats retain some Ran-dependent nuclear localization elements (*Figure 2—figure supplement 1*). Future studies will need to define the NLSs in the N-terminus and ARM repeats and the associated NTRs.

For the purposes of rational drug design, we fortuitously identified the β-catenin NTR as TNPO1, as it is one of the few NTRs where the NLS-NTR interaction is resolved to the atomic level (*Cansizoglu et al., 2007*; *Soniat et al., 2013*). This knowledge base has established a consensus amino acid sequence (PY-NLS) that helped us identify the β-catenin NLS required for TNPO1 binding. Furthermore, it has led to the generation of the high affinity M9M peptide (*Cansizoglu et al., 2007*) that, as shown here, can be used to inhibit Wnt signaling in TCF/LEF luciferase mouse fibroblast cell lines, demonstrating proof of principle that inhibiting TNPO1 could be a viable therapeutic strategy. Nonetheless, such a strategy might benefit from future crystallographic studies of the TNPO1-β-catenin complex, as the overall binding of different NLSs can vary depending on the contribution of individual amino acids. It may be possible, for example, to identify small molecules that specifically block the β-catenin-TNPO1 interaction without more broadly impacting other TNPO1 cargos. Our findings have formed the basis for ongoing work seeking to identify small molecules that could specifically target and ameliorate the multitude of Wnt-related diseases including cancers.

## Materials and methods
### Contact for reagent and resource sharing

Further information and request for reagents may be directed to and will be fulfilled by the lead contact, Mustafa K. Khokha (Mustafa.khokha@yale.edu).

### *Xenopus*

*X. tropicalis* and *Xenopus laevis* were maintained and cared for in accordance with the Yale University Institutional Animal Care and Use Committee protocols. In vitro fertilization was performed as per standard protocols (*del Viso and Khokha, 2012*; *Sive et al., 2007*).

### *S. cerevisiae* strains

All yeast strains used in this study are listed in *Supplementary file 1*. Yeast strains were grown at 30°C, unless indicated otherwise in yeast extract peptone dextrose (YPD) medium (1% bacto yeast extract,

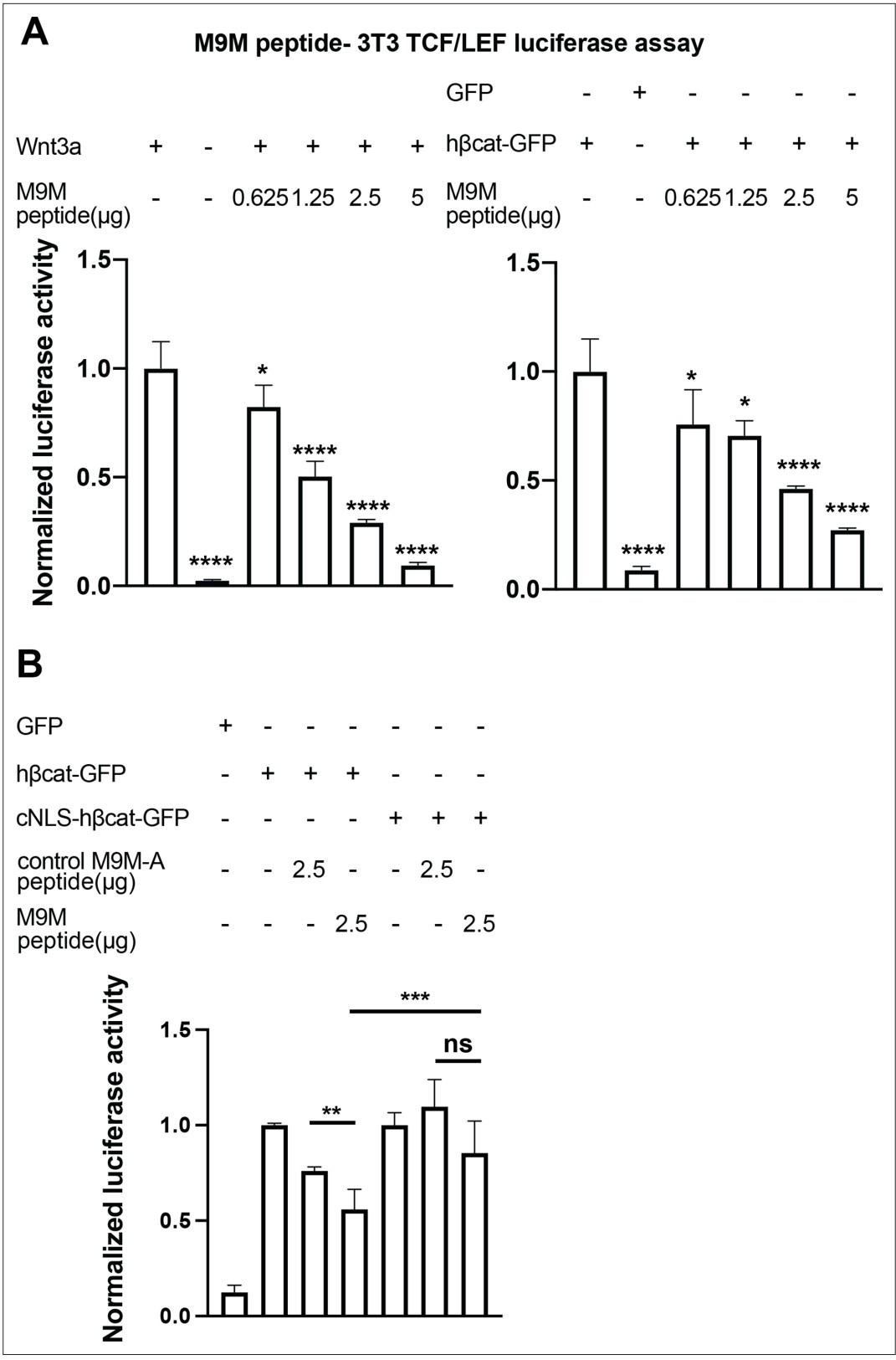

**Figure 7.** The M9M peptide inhibits Wnt signaling. Wnt signaling was activated by Wnt3a (**A**, left), human β-catenin-GFP overexpression (**A**, right and B), or cNLS-human β-catenin-GFP (**B**). No Wnt3a or GFP overexpression were used as negative controls. Experiments were performed in triplicate (**A**) or duplicate in two

*Figure 7 continued on next page*

*Figure 7 continued*

independent experiments (**B**). p-Values are from unpaired two-tailed t-test where ns is p>0.05, p<0.05 (*), 0.0021 (**), 0.0002 (***), and p<0.0001 (****). The data is uploaded as *Figure 7—source data 1*.

The online version of this article includes the following source data and figure supplement(s) for figure 7:

**Source data 1.** Source data related to *Figure 7*.

**Figure supplement 1.** Western blots of β-catenin from 3T3 TCF/LEF luciferase assays with M9M peptide treatment.

2% bactopeptone, 2% glucose, and 0.05% adenine sulfate). Transformation of yeast was carried out using standard protocols (*Amberg and Strathern, 2005*).

## Mammalian cells

HEK293T and HeLa cells were maintained and cultured with Dulbecco's Modified Eagle Medium (DMEM) medium + 10% fetal bovine serum + 1% penicillin and streptomycin in a T-75 flask. The engineered 3T3 mouse fibroblast cell line that stably expresses the TCF/LEF luciferase transgene was maintained with DMEM medium + Cell Growth Medium Concentrate (Enzo life sciences). Further experimental procedures for the 3T3 cell line can be found under TCF/LEF luciferase assay method section. Upon 70–80% cell confluency, cells were transfected with plasmids using jetPRIME (Polyplus-transfection) following the manufacturer's instructions. Cells were fixed with 4% paraformaldehyde/PBS and further processed for immunofluorescence 24–48 hr post-transfection. Antibodies used in this study are listed in Key resources table. Cells were tested for mycoplasma using MycoAlert Detection Kit (Lonza).

## Plasmid, mRNA, siRNA, CRISPR, and M9M peptide

Key resources and all plasmids used in this study are listed in Key resources table and *Supplementary file 2*, respectively. *Xenopus* β-catenin-GFP (Addgene #16839), *Xenopus* cNLS-β-catenin-GFP (Addgene #16838), GST-human β-catenin (Addgene #24193), and NLS-mCherry (Addgene #49313) plasmids were obtained from Addgene. GST-transportin 1 (TNPO1) and Gal-MBP(x3)-GFP plasmids were generous gifts from Dr. Yuh-min Chook at UTSW and Dr. Liesbeth M. Veenhoff at University of Groningen, respectively. Gibson Assembly (New England Biolabs) was used to generate GFP-tagged *Xenopus* β-catenin truncation constructs following the manufacturer's instructions. Both the human and *Xenopus* β-catenin P687A, M688A variants were generated using Q5 site- directed mutagenesis (New England Biolabs) following the manufacturer's instructions. Subsequently, the β-catenin constructs were sub-cloned into the pRS406 vector containing an ADH1 promotor for yeast studies or a pCS2+ vector for mammalian/*Xenopus* studies. mRNAs were generated using the SP6 mMessage machine kit (Thermo Fisher Scientific) and RNA clean & concentrator kit (Zymo Research) following the manufacturer's instructions. We obtained siRNAs directed against mouse TNPO1 (s108857), mouse TNPO2 (s102754), and a control siRNA (4390843) from Thermo Fisher Scientific. To generate CRISPR sgRNAs, we used the EnGen sgRNA synthesis kit (NEB) following the manufacturer's instructions with the following targeting sequences *tnpo1* sgRNA#1 (5'-GGCATGGGGGCCACCTCTTG-3'), *tnpo1* sgRNA#2 (5'- GGGTTACGTTTGTCCTCAAG-3'), *tnpo2* sgRNA #1 (5'- GGGCGTTTAGCCGCGT TCTA-3'), and *tnpo2* sgRNA #2 (5'- GGCGTCATGGATGAGTCCGA-3') (designed using CRISPRscan *Moreno-Mateos et al., 2015*). CRISPR experiments in wildtype or transgenic *X. tropicalis* were performed as previously described (*Bhattacharya et al., 2015*). CRISPR gene editing efficiency was assessed using Synthego ICE (ice.synthego.com) as previously described (*Sempou et al., 2018*). The M9M peptide (GGSYNDFGNYNNQSSNFGPMKGGNFGGRFEPYANPTKR) and the M9M-A peptide ( GGSYNDFGNYNNQSSNAAAAKGGNFGGAFEAAANPTKR) were synthesized by LifeTein.

## Yeast and mammalian β-catenin sub-cellular localization by microscopy

Expression plasmids containing full length and fragments of the β-catenin-GFP coding sequence under the control of the *ADH1* promoter were transformed into the W303, Heh2-mCherry::NAT (BWCPL1314) strain. A yeast colony that incorporated the plasmid sequence was cultured and mounted onto a coverslip for live imaging on a DeltaVision wide-field microscope (GE Healthcare) with a CoolSnapHQ$^2$ CCD camera. Yeast fluorescent images were deconvolved using the iterative algorithm sofWoRx. 6.5.1

(Applied Precision, GE Healthcare). β-catenin-GFP transfected HeLa or HEK293T cells were mounted on Pro-Long Gold coated coverslip for imaging on a Zeiss Axio Observer microscope. All fluorescent images were analyzed with Fiji software. For quantification, the oval selection tool was used to draw a circle in both the nuclear and cytosolic regions on the same image plane to measure florescence intensity.

## Secondary axis assays and β-catenin sub-cellular localization

*X. laevis* embryos were injected with a mixture of either 200 pg of *Xenopus* or human β-catenin-GFP mRNA and cNLS-mCherry mRNA in one of four cells (targeting the ventral side). Embryos were assessed for a secondary axis via stereomicroscopy at stage 17–19. For the β-catenin localization experiments by fluorescence, stage 10 embryos were fixed in 4% paraformaldehyde/PBS at 4°C overnight on a nutator. Embryos were washed in 1× PBS + 0.1% TritonX-100, and the dorsal blastopore lips were sectioned with a razor blade and mounted on Pro-Long Gold (Invitrogen) coated coverslip before imaging on a Zeiss 710 confocal microscope.

## The Anchor-Away assay

To employ the Anchor-Away approach, we used a yeast strain that harbors a FKBP12 fusion of the endogenous plasma membrane $H^+$-ATPase (*PMA1* gene), a Heh2-mCherry fusion to mark the nuclear envelope and a mutated *TOR1* gene (HHY110: *HEH2-mCherry::KAN, PMA1-2xFKBP12, fpr1::NAT tor1-1*). In this strain, individual, endogenous NTRs are tagged with the FRB domain at the C-terminus by homologous recombination of a PCR product that contains an FRB sequence, a 3× HA epitope, and a selective marker, *HIS3*, flanked by a 60 bp homology arm of endogenous NTR coding sequence (*Figure 3—figure supplement 2* and *Supplementary file 3* for FRB cloning primers; *Haruki et al., 2008*; *Longtine et al., 1998*). Integration of an FRB sequence is confirmed by colony PCR using a gene-specific forward and a plasmid specific reverse primer and rapamycin-induced cell death for essential NTRs (*Figure 3—figure supplement 2* and *Supplementary file 3* for colony PCR primers). Subsequently, an expression plasmid containing the coding sequence for xβ-catenin (665-782)-GFP under the control of the *ADH1* promoter was transformed into the Anchor-Away line. These lines were treated with 1 mg/ml of rapamycin (5–15 min of incubation) or vehicle alone (DMSO) before imaging.

## TCF/LEF luciferase assay

A 3T3 mouse fibroblast cell line that has a stable integration of the luciferase reporter gene under the Wnt responsive TCF/LEF promoters was used for this assay (Enzo life sciences). Cells were maintained with DMEM medium + Cell Growth Medium Concentrate (Enzo life sciences). Prior to the transfection, cells were seeded on a 24 well plate in media containing DMEM + Cell Assay Medium Concentrate (Enzo life sciences). Subsequently, cells were transfected with siRNA (25 pmol) first, then GFP or human β-catenin-GFP DNA (0.5 µg) for 48 hr and 24 hr, respectively, using JetPRIME per the manufacturer's instructions. Luciferase was quantified using the Luciferase Assay System (Promega) and the Promega Glomax luminometer according to the manufacturer's instructions. For the M9M peptide experiment, cells were transfected with an M9M or M9M-A peptide dose ranging from 0.635 µg to 0.5 µg using ProteoJuice Protein transfection following the manufacturer's instructions for 20 hr, and Wnt signaling was activated either by Wnt3a ligand (50 ng/ml) or human β-catenin-GFP DNA (0.5 µg) for 16–24 hr.

## In vivo TCF/LEF GFP in situ hybridization

Heterozygous *X. tropicalis* Tg(*pbin7Lef-dGFP*) were crossed with wildtype *X. tropicalis*. Fertilized embryos were injected with sgRNAs targeting *tnpo1* and *tnpo2* and Cas9 protein at one-cell stage and collected at stage 10 for in situ hybridization as previously described (*Khokha et al., 2002*). Digoxigenin-labeled anti-sense GFP probe was used to detect GFP transcript expression. Progeny that did not carry the transgene was used as a negative control.

## Western blotting

3T3 mouse fibroblast cells were lysed in radioimmunoprecipitation assay (RIPA) buffer to harvest protein samples. Nuclear and cytoplasmic proteins were extracted from colorectal cancer cell lines (HCT-116 and DLD-1) following the manufacturer's instructions (NE-PER Thermo Fisher). Protein levels

are normalized, and immunoblots were carried out in bolt 4–12% Bis-Tris plus gels following standard protocols. Antibodies used in this study are listed in Key resources table.

## In vitro binding experiment

pGEX-6P1, pGEX-human β-catenin, pGEX-human transportin 1, or pET28a-His6-RanQ69L were transformed into the BL21 *Escherichia coli* strain and cultured in Lysogeny broth (LB) with antibiotics to mid-log phase ($OD_{600}$ 0.6–0.8). To induce expression of the recombinant proteins (GST alone, GST-hβ-catenin, GST-hTNPO1, and His6-hRanQ69L), isopropyl β-d-1-thiogalacto pyranoside was added at a final concentration of 1 mM for 3 hr. All cultures were harvested in 50 ml batches and stored at –80°C until further use. Glutathione Sepharose (GT) beads (Millipore Sigma) were washed and equilibrated in lysis buffer 1 (50 mM Tris pH 7.4, 150 mM NaCl, 2 mM $MgCl_2$, 10% glycerol, 0.05% NP-40, 1 mM DTT, and protease inhibitor cocktail mix [Millipore Sigma]). Bacterial pellets containing GST tagged proteins were resuspended with the ice cold lysis buffer 1, sonicated, and spun down at 4°C at 30,000× g for 20 min. The supernatant was collected into a new 50 ml conical tube and incubated with 200 µl of GT bead slurry for 1 hr at 4°C. Subsequently, the GST-GT bead slurry was collected and washed with lysis buffer 1 (excluding the protease cocktail mix). The GST tag was removed from hTNPO1 using proTEV Plus Protease (Promega), and the protease enzyme was further removed from hTNPO1 protein by Ni-NTA Magnetic Beads (NEB) as per the manufacturer's instructions. To pulldown His6-RanQ69L, bacterial pellets were resuspended with ice cold lysis buffer 2 (50mM Tris pH 7.4, 500 mM NaCl, 2 mM MgCl2, 20 mM Imidazole, 10% glycerol, 0.05% NP-40, and protease inhibitor cocktail mix), sonicated, and spun down at 4°C at 30,000× g for 20 min. The supernatant was collected into a new 50 ml conical tube and incubated with 200 µl of equilibrated Ni NTA bead slurry for 1 hr at 4°C. The His-Ni bead slurry was collected and washed with lysis buffer 2. His6-RanQ69L was eluted from Ni beads by adding the elution buffer (50 mM pH Tris 7.4, 500 mM NaCl, 2 mM MgCl2, 500 mM Imidazole, 10% glycerol, 0.05% NP-40, and protease inhibitor cocktail mix) and rotating at 4°C for 1 hr. The supernatant containing His-RanQ69L was collected by centrifugation at 500× g for 3 min and dialyzed in the buffer (50 mM pH Tris 7.4, 150 mM NaCl, 2 mM MgCl2, 10% Glycerol, and 1 mM PMSF) at 4°C overnight. Subsequently, RanQ69L was incubated in GTP buffer (400 uM GTP in 5 mM EDTA) 4°C for 1 hr. hNTPO1 protein was incubated with GT beads preloaded with GST fusion protein (GST alone or GST-hβ-catenin) for 1 hr at 4°C. Then, RanQ69L protein (dialysis buffer + GTP or RanQ69L) was added to the mix for 1 hr at 4°C. The beads were washed with the lysis buffer and eluted with SDS-PAGE sample buffer. Protein samples were separated by SDS-PAGE and detected with Coomassie (BioRad).

## Quantification and statistical analysis

Statistical significance was defined as $p < 0.05$ (*), 0.002 (**), 0.0002 (***), and 0.0001 (****). The double axis assay and in situ data were analyzed by Fisher's exact tests. Otherwise, unpaired two-tailed Student's t-tests or a one-way ANOVA test were used to determine significance of mean ratio of nuclear to cytosolic fluorescence intensity in GraphPad Prism 8.4.3.

## Acknowledgements

We thank M Slocum and M Lane for *Xenopus* husbandry and E Rodriguez, N Ader, and S Chandra for technical assistance and advice. For providing plasmids, we thank Dr. Yuh Min Chook at UTSW and Addgene. We thank the National *Xenopus* Resource at the Marine Biological Laboratory for distributing the *X. tropicalis Tg(pbin7Lef-dGFP)* line. We thank the Yale Center for Advanced Light Microscopy for their assistance with confocal imaging. WYH was supported by the Paul and Daisy Soros Fellowship for New Americans and the NIH (5F30HL143878). WYH and VK were supported by the Yale MSTP NIH T32GM07205. DPG was supported by the NIH (5F31HL149246). MKK and CPL were supported by the NIH (2R01HL124402).

# Additional information

### Competing interests

Mustafa K Khokha: is a co-founder of Victory Genomics, Inc. The other authors declare that no competing interests exist.

### Funding

| Funder | Grant reference number | Author |
|---|---|---|
| National Institutes of Health | 2R01HL124402 | C Patrick Lusk<br>Mustafa K Khokha |
| National Institutes of Health | T32GM07205 | Woong Y Hwang<br>Valentyna Kostiuk |
| National Institutes of Health | 5F30HL143878 | Woong Y Hwang |
| National Institutes of Health | 5F31HL149246 | Delfina P González |
| Paul and Daisy Soros Fellowships for New Americans | | Woong Y Hwang |

The funders had no role in study design, data collection and interpretation, or the decision to submit the work for publication.

### Author contributions

Woong Y Hwang, Conceptualization, Data curation, Formal analysis, Validation, Investigation, Visualization, Methodology, Writing - original draft, Writing – review and editing; Valentyna Kostiuk, Delfina P González, Investigation; C Patrick Lusk, Conceptualization, Supervision, Funding acquisition, Methodology, Writing – review and editing; Mustafa K Khokha, Conceptualization, Supervision, Funding acquisition, Writing – review and editing

### Author ORCIDs

Woong Y Hwang ⓘ http://orcid.org/0000-0002-0575-0033
Valentyna Kostiuk ⓘ http://orcid.org/0000-0003-3405-7518
Delfina P González ⓘ http://orcid.org/0000-0002-6327-1348
C Patrick Lusk ⓘ http://orcid.org/0000-0003-4703-0533
Mustafa K Khokha ⓘ http://orcid.org/0000-0002-9846-7076

### Ethics

Xenopus tropicalis and Xenopus laevis were housed and cared for in our aquatics facility according to established protocols approved by the Yale Institutional Animal Care and Use Committee (IACUC, protocol number-2021-11035).

### Decision letter and Author response

Decision letter https://doi.org/10.7554/eLife.70495.sa1
Author response https://doi.org/10.7554/eLife.70495.sa2

# Additional files

### Supplementary files

• Supplementary file 1. List of genotypes and origins of all *Saccharomyces cerevisiae* strains used in this study.

• Supplementary file 2. List of all plasmids used in this study.

• Supplementary file 3. List of primers used for the Anchor-Away assay in this study.

• Transparent reporting form

• Source data 1. Source data related to original gels and western blots.

## Data availability

Data generated or analyzed during this study are included in the manuscript and the supporting file. Source data file has been provided for Figure 1, 2B, 3C, 4B, 4C, 6B, 7, Figure 2-figure supplement 1, Figure 2- figure supplement 2.

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

# Appendix 1

## Appendix 1—key resources table

| Reagent type (species) or resource | Designation | Source or reference | Identifiers | Additional information |
|---|---|---|---|---|
| Strain and strain background (*Xenopus tropicalis*) | Tg(pbin7LEF-dGFP) | National *Xenopus* Resources at MBL | NXR_1094 | |
| Strain and strain background (*Xenopus laevis*) | *X. laevis* | NASCO | LM00535 and LM00715 | |
| Strain and strain background (*Escherichia coli*) | BL21 Gold (DE3) | Agilent | 230132 | |
| Strain and strain background (*E. coli*) | XL-10 Gold | Agilent | 200314 | |
| Strain and strain background (*E. coli*) | DH5-alpha | NEB | C2987 | |
| Cell line (*M. musculus*) | Leading Light Wnt Reporter Cell line-TCF/ LEF luciferase 3T3 mouse fibroblast | Enzo Life Sciences | ENZ-61001–0001 | |
| Cell line (*Homo-sapiens*) | Human embryonic kidney 293 (HEK293T) | ATCC | CRL-3216 | |
| Cell line (*Homo-sapiens*) | HeLa | ATCC | CCL-2 | |
| Cell line (*Homo-sapiens*) | Human colorectal cancer (HCT 116) | ATCC | CCL-247 | |
| Cell line (*Homo-sapiens*) | Human colorectal cancer (DLD-1) | ATCC | CCL-221 | |
| Transfected construct (*M. musculus* and human) | siRNA to TNPO1 & 2 | Thermo Fisher | | |
| Antibody | Anti-β-catenin (mouse monoclonal) | Santa Cruz | sc-7963 HRP, RRID:AB_626807 | WB (1:1000) |
| Antibody | Anti-β-actin (mouse monoclonal) | Santa Cruz | sc-47778 HRP, RRID:AB_2714189 | WB (1:10000) |
| Antibody | Anti-GFP (mouse monoclonal) | Santa Cruz | sc-9996 HRP, RRID:AB_627695 | WB (1:1000) |
| Antibody | Anti-Transportin-1 (mouse monoclonal) | Abcam | ab10303, RRID:AB_2206878 | WB (1:1000) |
| Antibody | Anti-Transportin-2 (Rabbit polyclonal) | Proteintech | 17831–1-AP, RRID:AB_10598481 | WB (1:3000) |
| Antibody | Anti-LaminB1 (Rabbit polyclonal) | Abcam | Ab16048, RRID:AB_443298 | IF (1:500) WB (1:1000) |
| Antibody | Anti-GAPDH (mouse monoclonal) | Santa Cruz | sc-47724 HRP; RRID:AB_627678 | WB (1:3000) |
| Sequence-based reagent | tnpo1 CRISPR 1 | This paper | Oligonucleotides | ttctaatacgactcactata GGCATGGGGGCCACCTCTTG gttttagagctagaa |
| Sequence-based reagent | tnpo1 CRISPR 2 | This paper | Oligonucleotides | ttctaatacgactcactata GGGTTACGTTTGTCCTCAAG gttttagagctagaa |
| Sequence-based reagent | tnpo2 CRISPR 1 | This paper | Oligonucleotides | ttctaatacgactcactata GGGCGTTTAGCCGCGTTCTA gttttagagctagaa |
| Sequence-based reagent | tnpo2 CRISPR 2 | This paper | Oligonucleotides | ttctaatacgactcactata GGCGTCATGGATGAGTCCGA gttttagagctagaa |
| Sequence-based reagent | siRNA: negative control | Thermo Fisher | 4390843 | Silencer Select |
| Sequence-based reagent | siRNA: mouse TNPO1 | Thermo Fisher | s108857 | Silencer Select |
| Sequence-based reagent | siRNA: mouse TNPO2 | Thermo Fisher | s102754 | Silencer Select |
| Sequence-based reagent | siRNA: human TNPO1 | Thermo Fisher | s7934 | Silencer Select |
| Sequence-based reagent | siRNA: human TNPO2 | Thermo Fisher | s26881 | Silencer Select |

*Appendix 1 Continued on next page*

*Appendix 1 Continued*

| Reagent type (species) or resource | Designation | Source or reference | Identifiers | Additional information |
|---|---|---|---|---|
| Peptide, recombinant protein | M9M-A | LifeTein | Custom | N-GGSYNDFGNYNNQSSNAAA AKGGNFGGAFEAAANPTKR-C |
| Peptide, recombinant protein | M9M | LifeTein | Custom | N-GGSYNDFGNYNNQSSNFGPMK GGNFGGRFEPYANPTKR-C |
| Commercial assay or kit | Luciferase Assay System | Promega | E1500 | |
| Commercial assay or kit | NE-PER Nuclear Cytoplasmic Extraction Reagents | Thermo Scientific | 78833 | |
| Commercial assay or kit | jetPRIME | Polyplus transfection | 114–15 | |
| Commercial assay or kit | ProteoJuice Protein Transfection | Millipore Sigma | 71281 | |
| Commercial assay or kit | MycoAlert Detection Kit | Lonza | LT07-118 | |
| Chemical compound and drug | Rapamycin | Fisher scientific | AAJ62473MF | |
| Chemical compound and drug | Glutathione Sepharose 4B | Millipore Sigma | GE17-0756-01 | |
| Chemical compound and drug | Protease Inhibitor Cocktail mix | Millipore Sigma | P8340-5ML | |
| Chemical compound and drug | ProTEV Plus | Promega | V6101 | |
| Chemical compound and drug | NEBExpress Ni-NTA Magnetic Beads | NEB | S1423S | |
| Chemical compound and drug | Isopropyl β-d-1-thiogalacto pyranoside (IPTG) | Thermo Fisher | 15529019 | |
| Software and algorithm | Fiji | ImageJ | https://imagej.net/Fiji | |
| Software and algorithm | Prism 9 | Graphpad | https://www.graphpad.com/ | |

