## [Editor Report]

In Wnt/β-catenin signaling, Wnt growth factor binding to cell surface receptors results in the stabilization and nuclear translocation of the transcriptional coactivator β-catenin, but how β-catenin is translocated to the nucleus has been a longstanding problem in the field. The authors show that the yeast Kap104/mammalian TNPO1 mediates nuclear translocation of β-catenin using a conserved TNPO1 nuclear localization sequence in the C-terminal region of β-catenin, and mutation of this sequence or knockdown of TNPO1 diminishes nuclear localization and Wnt sigaling. The data demonstrate that β-catenin nuclear translocation is Ran dependent and that TNPO1 binding is a significant, although not exclusive, contributor to β-catenin translocation, and could represent a new therapeutic target.

---

## [Decision Letter]

**Decision letter after peer review:**

Thank you for submitting your article "Kap-β2/Transportin mediates β-catenin nuclear transport in Wnt signaling" for consideration by *eLife*. Your article has been reviewed by 3 peer reviewers, and the evaluation has been overseen by a Reviewing Editor and Kevin Struhl as the Senior Editor. The following individual involved in review of your submission has agreed to reveal their identity: Yuh Min Chook (Reviewer #1).

Essential revisions:

The reviewers agree that the findings here are significant and contribute to our understanding of β-catenin signaling. However, there were several problems that need to be addressed, including the constructs used for assessing nuclear translocation of β-catenin fragments, the pulldown assay, and a general concern about TNPO1 as a therapeutic target.

1. The fluorescence data show significant nuclear localization of β-catenin even in the absence of TNPO1 activity. Thus, there may be both Ran-dependent and -independent mechanisms for its nuclear localization, and this should be made clear, especially since there are prior findings along these lines. Given this, it is premature to claim that TNPO1 would be a target for inhibiting β-catenin signaling in cancer if it is only 50% effective. Double knockdown of TNPO1/2 only decreased WNT reporter activity by 60%. Β-catenin P687A M688A mutant still retained significant activity in the *Xenopus* double axis assay (Figure 5C). Further, a significant fraction of β-catenin Δ 665-745-GFP was in the nucleus (Figure S2C). If TNPO1/2 are only partially responsible for β-catenin nuclear translocation, targeting the TNPO1/2-β-catenin interaction might not be a good therapeutic strategy for cancer. This can be provided as a speculative discussion point, but not a conclusion of the paper.

2. Related to (1), inhibition of TNPO1/2-dependent nuclear import pathway was not confirmed in a relevant (e.g. colorectal) cancer model system, in which β-catenin levels are inappropriately enhanced and inhibition of β-catenin nuclear entry would needed to support the idea that the current findings hold relevance for the development of future cancer therapies. The importance of TNPO1/2 for nuclear translocation of β-catenin should be tested in more mammalian cell lines, especially since a IPO11-β-catenin connection has already been proposed. TNPO1/2 knockdown and/or β-catenin P687A M688A viral transduction in various cell lines, including cell lines with β-catenin or APC mutations, and nuclear β-catenin levels quantified. Does the MPM9 peptide and/or TNPO1/2 depletion inhibit β-catenin mediated transcription in cancer cell lines carrying either β-catenin or APC mutations?

3. The difference between β-catenin-GFP and β-catenin δ 665-745 is quite clear in *Xenopus* embryos (Figure S2C). It would be more convincing if authors can extend this finding to common mammalian cell lines such as HEK293, Hela or colon cancer cell lines such as DLD1. This can be done by lentiviral transduction and should be a very easy experiment for them.

4. Since GFP is small and can enter the nucleus passively, it might not be a good fusion partner for mapping nuclear localization activity of β-catenin truncation fragments (Figure 2B). Based on Figure 2B, it appears that many fragments of β-catenin have some level of NLS activity. It would be more convincing if authors can fuse different β-catenin fragments to MBP(3x)GFP, which is excluded from the nucleus (Figure 4C). A control with an established TNPO1 cargo like hnRNP A1 or a strong YP-NLS would be helpful as well. It is also not clear whether the authors used MBP(3x)GFP to test the activity of the PY like motif in HeLa cells (Figure 4D); if not they should do so.

5. The pull-down binding assay of GST-β-catenin + TNPO1 in Figure 4B could benefit from some straightforward improvement. Lane 4 shows TNPO1 binding, but at very sub-stoichiometric amounts. This may be due to having partially active bacterially expressed GST- TNPO1, which is not easy to produce. While there appears to be a decrease in binding to the PM/AA mutant, the experiment would be more decisive if RanGTP is added to the pulldown in lane 4 to demonstrate specific and Ran-sensitive TNPO1-β-catenin interactions.

6. It is possible that TNPO1/2 are required for nuclear translocation of cofactors of β-catenin; the C-terminus of β-catenin is involved in transcription activation, and it is thought to bind to many nuclear factors. Can the authors rule this out?

Other changes for the introduction/discussion:

a. The authors are correct that NTRs are mostly functionally conserved between the budding yeast and mammalian systems. However, PY-NLS/cargo binding functions of Kap104 are only partially conserved when compared to those of TNPO1. Since the key finding here is on Kap104 and TNPO1, the authors should present this information more carefully and thoroughly in the introduction. Please refer to studies of Kap104 binding to the PY-NLS and conservation of PY-NLS recognition across eukaryotic TNPO1 in Suel et al., PLoS Biology 2008.

TNPO1 binds to PY-NLSs that carry either basic (e.g. hnRNP M) or hydrophobic (e.g. hnRNP A1 and FUS) N-terminal motifs/epitopes. Please note that Kap104 binds only PY-NLSs that carry the basic epitope (e.g. Nab2 and as the authors show here, β-catenin) but not the ones that have hydrophobic N-terminal epitopes (hnRNP A1, FUS, etc. hence the use of budding yeast to study toxicity of cytoplasmic FUS). This is also why M9M (has an N-terminal hydrophobic epitope) works in mammalian cells but not in yeast.

b. It is suggested that the authors simply use the term PY-NLS, which is defined very broadly (see the Soniat and Chook 2015 review and other reviews on nuclear transport and NLSs), for the NLS of β-catenin. Sequences of PY-NLSs are incredibly diverse and defined only by very loose sequence motifs, and this class of signals is simply named after the most easily recognized and most conserved PY sequence element. The Suel et al., paper shows clearly how other hydrophobic residues function well in the place of the tyrosine in the PY motif of this signal. For example, the yeast mRNA export factor Nab2 carries a PY-NLS that has a PL motif. Soniat et al., (Structure 2016) also showed how a PY-NLS can sometimes not even have a PY motif, and the 2008 Suel et al., paper explained why this is the case.

c. The PY dipeptide motif is simply the most easily recognized and most conserved PY sequence element. It is NOT the most energetically important element of the PY-NLS; it is just one of several binding elements usually present in the signal so the PM/PA mutation should not be expected to completely inactivate the NLS let alone nuclear localization. The PY is in fact weak (in the extreme, missing) and does not contribute much to TNPO1 or Kap104 binding in some PY-NLSs. An example is the M9 sequence of the PY-NLS of hnRNP A1, which has a really weak PY motif (Lee et al., Cell 2006).

[Editors’ note: further revisions were suggested prior to acceptance, as described below.]

Thank you for resubmitting your work entitled "Kap-β2/Transportin mediates β-catenin nuclear transport in Wnt signaling" for further consideration by *eLife*. Your revised article has been evaluated by Kevin Struhl (Senior Editor) and a Reviewing Editor.

The manuscript has been improved but there are some remaining issues that need to be addressed, as outlined below:

1. Given the emphasis on the potential therapeutic importance, the concern remains that the requirement of this NLS for β-catenin localization in mammalian cells is not solidly established. The TNPO knockdown experiment has many caveats (off target, indirect effect through other proteins). Therefore, it was felt that an experiment to test whether a β-catenin mutant lacking this NLS has decreased nuclear localization in mammalian cells should be performed. Note that even if the result is negative, you can discuss the caveats.

2. The knockdown experiments in the HCT116 and DLD1 cells show a weak effect (Figure 6-supplement 4) – is it possible to show longer exposures?

---

## [Author Response]

1. The fluorescence data show significant nuclear localization of β-catenin even in the absence of TNPO1 activity. Thus, there may be both Ran-dependent and -independent mechanisms for its nuclear localization, and this should be made clear, especially since there are prior findings along these lines.

The reviewer raises an important point, and we agree that there may be other elements beyond the TNPO1 NLS that are required for b-catenin nuclear entry; however, our data argue against any Ran-independent mechanism. For example, we now include new data examining the localization of aa 1-664-GFP (containing the N-terminus and ARM repeats but lacking the TNPO1 NLS), in the temperature sensitive Ran-GEF strain (mtr1-1). As shown in Figure 2—figure supplement 3, the residual nuclear accumulation of 1-664-GFP is in fact dependent on a functional Ran pathway. Thus, as is now more fully addressed in the revised discussion, although the TNPO1 NLS is the “strongest” nuclear localization element, it is possible that there are other NLSs in the rest of the protein that await future characterization.

Given this, it is premature to claim that TNPO1 would be a target for inhibiting β-catenin signaling in cancer if it is only 50% effective. Double knockdown of TNPO1/2 only decreased WNT reporter activity by 60%. Β-catenin P687A M688A mutant still retained significant activity in the *Xenopus* double axis assay (Figure 5C). Further, a significant fraction of β-catenin Δ 665-745-GFP was in the nucleus (Figure S2C). If TNPO1/2 are only partially responsible for β-catenin nuclear translocation, targeting the TNPO1/2-β-catenin interaction might not be a good therapeutic strategy for cancer. This can be provided as a speculative discussion point, but not a conclusion of the paper.

Certainly, until it is tested, we will not know if inhibiting the TNPO1-β-catenin interaction is a good therapeutic strategy for cancer. We have been more explicit about this point in the discussion.

2. Related to (1), inhibition of TNPO1/2-dependent nuclear import pathway was not confirmed in a relevant (e.g. colorectal) cancer model system, in which β-catenin levels are inappropriately enhanced and inhibition of β-catenin nuclear entry would needed to support the idea that the current findings hold relevance for the development of future cancer therapies. The importance of TNPO1/2 for nuclear translocation of β-catenin should be tested in more mammalian cell lines, especially since a IPO11-β-catenin connection has already been proposed. TNPO1/2 knockdown and/or β-catenin P687A M688A viral transduction in various cell lines, including cell lines with β-catenin or APC mutations, and nuclear β-catenin levels quantified. Does the MPM9 peptide and/or TNPO1/2 depletion inhibit β-catenin mediated transcription in cancer cell lines carrying either β-catenin or APC mutations?

As suggested, we depleted TNPO1/2 by siRNA in two different colorectal cell lines (HCT-116 and DLD-1) and showed a significant reduction of β-catenin protein levels in the nucleus compared to the control siRNA condition by western blot. We have added this data to Figure 6—figure supplement 4.

3. The difference between β-catenin-GFP and β-catenin δ 665-745 is quite clear in *Xenopus embryos* (Figure S2C). It would be more convincing if authors can extend this finding to common mammalian cell lines such as HEK293, Hela or colon cancer cell lines such as DLD1. This can be done by lentiviral transduction and should be a very easy experiment for them.

In Figure 2 figure supplement 1, we test xbeta-catenin-(665-745)-GFP in HEK293T cells which enriches in the nucleus to a much greater degree than GFP alone. Figure S2C is supportive data for the secondary axis experiments in *Xenopus*. We felt it was important to show that β-catenin-GFP localization in the *Xenopus* context also correlates with the incidence of secondary axes. Then in HeLa cells, we show that β-catenin 665-782-GFP enriches in the nucleus, which does not occur when we mutate the PM residues to AA (Figure 4C). Then we demonstrate that this point mutant leads to a loss of direct binding between β-catenin and TNPO1 (this interaction is between human proteins) (Figure 5A). Therefore, both the in vitro binding experiment and the localization of these constructs in mammalian cells provides strong evidence of the NLS’s functionality in the mammalian system.

4. Since GFP is small and can enter the nucleus passively, it might not be a good fusion partner for mapping nuclear localization activity of β-catenin truncation fragments (Figure 2B). Based on Figure 2B, it appears that many fragments of β-catenin have some level of NLS activity. It would be more convincing if authors can fuse different β-catenin fragments to MBP(3x)GFP, which is excluded from the nucleus (Figure 4C). A control with an established TNPO1 cargo like hnRNP A1 or a strong YP-NLS would be helpful as well. It is also not clear whether the authors used MBP(3x)GFP to test the activity of the PY like motif in HeLa cells (Figure 4D); if not they should do so.

We agree that fragments outside the C-terminus of β-catenin are likely to contain an NLS element. We tested a C-terminal deletion and demonstrate that the nuclear enrichment is lost at the non-permissive temperature in the mtr1-1 strain indicating Ran dependence (New Data – Figure 2—figure supplement 3). In this manuscript, we focused on the C-terminus as it was required for the most significant fraction of the nuclear enrichment. Future studies can investigate the necessary NTRs in the ARM and N-terminus.

As the reviewer suggests, we tried to localize MBP(3x)-GFP in HeLa cells. However, as show in Author response image 1, the MBP(3x)-GFP aggregated in the cytosol in these cell lines precluding the ability to perform the analysis as suggested.

**Author response image 1. sa2fig1:** 

5. The pull-down binding assay of GST-β-catenin + TNPO1 in Figure 4B could benefit from some straightforward improvement. Lane 4 shows TNPO1 binding, but at very sub-stoichiometric amounts. This may be due to having partially active bacterially expressed GST- TNPO1, which is not easy to produce. While there appears to be a decrease in binding to the PM/AA mutant, the experiment would be more decisive if RanGTP is added to the pulldown in lane 4 to demonstrate specific and Ran-sensitive TNPO1-β-catenin interactions.

As requested, we purified a his-tagged RanQ69L protein (ie GTP hydrolysis deficient Ran mutant), loaded it with GTP and showed that RanQ69L-GTP selectively forms complexes with TNPO1 and disrupts the TNPO1-β-catenin interaction. We have added this data to Figure 5 and Figure 5—figure supplement 1.

6. It is possible that TNPO1/2 are required for nuclear translocation of cofactors of β-catenin; the C-terminus of β-catenin is involved in transcription activation, and it is thought to bind to many nuclear factors. Can the authors rule this out?

We have met the standard burden for defining whether a given protein is a transport cargo of an NTR: we define, in multiple systems including a minimal budding yeast model that lacks the majority of physiological β-catenin transcriptional activators, that TNPO1 is required for its nuclear import. We map the TNPO1 binding site to a minimal sequence that resembles established TNPO1 NLSs (demonstrate the sequence is necessary and sufficient for nuclear import in multiple model systems) and then show that this sequence is required for direct binding of TNPO1 to β-catenin via in vitro assays with recombinant proteins. Further, to solidify it is a canonical NTR-NLS interaction, we also demonstrate that the TPNO1-β-catenin complex is disrupted by Ran-GTP. Whether TNPO1 also transports co-factors of β-catenin that are required for its role as a transcriptional activator is certainly possible, but the sum of our data indicate that TNPO1 binds directly to β-catenin for nuclear transport.

Other changes for the introduction/discussion:a. The authors are correct that NTRs are mostly functionally conserved between the budding yeast and mammalian systems. However, PY-NLS/cargo binding functions of Kap104 are only partially conserved when compared to those of TNPO1. Since the key finding here is on Kap104 and TNPO1, the authors should present this information more carefully and thoroughly in the introduction. Please refer to studies of Kap104 binding to the PY-NLS and conservation of PY-NLS recognition across eukaryotic TNPO1 in Suel et al., PLoS Biology 2008.

We appreciate this point and regret our oversimplification. We have addressed this with proper citations in the discussion.

TNPO1 binds to PY-NLSs that carry either basic (e.g. hnRNP M) or hydrophobic (e.g. hnRNP A1 and FUS) N-terminal motifs/epitopes. Please note that Kap104 binds only PY-NLSs that carry the basic epitope (e.g. Nab2 and as the authors show here, β-catenin) but not the ones that have hydrophobic N-terminal epitopes (hnRNP A1, FUS, etc. hence the use of budding yeast to study toxicity of cytoplasmic FUS). This is also why M9M (has an N-terminal hydrophobic epitope) works in mammalian cells but not in yeast.

We thank the reviewer for this insightful information. We have incorporated a discussion of this point in the main text.

b. It is suggested that the authors simply use the term PY-NLS, which is defined very broadly (see the Soniat and Chook 2015 review and other reviews on nuclear transport and NLSs), for the NLS of β-catenin. Sequences of PY-NLSs are incredibly diverse and defined only by very loose sequence motifs, and this class of signals is simply named after the most easily recognized and most conserved PY sequence element. The Suel et al., paper shows clearly how other hydrophobic residues function well in the place of the tyrosine in the PY motif of this signal. For example, the yeast mRNA export factor Nab2 carries a PY-NLS that has a PL motif. Soniat et al., (Structure 2016) also showed how a PY-NLS can sometimes not even have a PY motif, and the 2008 Suel et al., paper explained why this is the case.

As suggested, we will change to PY-NLS.

c. The PY dipeptide motif is simply the most easily recognized and most conserved PY sequence element. It is NOT the most energetically important element of the PY-NLS; it is just one of several binding elements usually present in the signal so the PM/PA mutation should not be expected to completely inactivate the NLS let alone nuclear localization. The PY is in fact weak (in the extreme, missing) and does not contribute much to TNPO1 or Kap104 binding in some PY-NLSs. An example is the M9 sequence of the PY-NLS of hnRNP A1, which has a really weak PY motif (Lee et al., Cell 2006).

Thank you for this information. We were fortunate that the mutation of the PM-AA disrupted the detectable binding of β-catenin to TPNO1 in vitro and in vivo.

[Editors’ note: further revisions were suggested prior to acceptance, as described below.]

1. Given the emphasis on the potential therapeutic importance, the concern remains that the requirement of this NLS for β-catenin localization in mammalian cells is not solidly established. The TNPO knockdown experiment has many caveats (off target, indirect effect through other proteins). Therefore, it was felt that an experiment to test whether a β-catenin mutant lacking this NLS has decreased nuclear localization in mammalian cells should be performed. Note that even if the result is negative, you can discuss the caveats.

To address this concern, we have generated a deletion in the NLS in human β-catenin-GFP. We have compared the localization of this protein with the wildtype β-catenin-GFP and demonstrate a significant difference. The wildtype β-catenin-GFP readily localizes to the nucleus of mammalian HEK293T cells while the nuclear localization DNLS is significantly less. These data are now included in a Figure 2 Supplement 2 panel C.

2. The knockdown experiments in the HCT116 and DLD1 cells show a weak effect (Figure 6-supplement 4) – is it possible to show longer exposures?

We regret that the exposure made the blot difficult to interpret. We have replaced this figure with a blot with a longer exposure which clarifies the results.